# Remodeling of whole-body lipid metabolism and a diabetic-like phenotype caused by loss of CDK1 and hepatocyte division

Jin Rong Ow[1], Matias J Caldez[1,2], Gözde Zafer[1,2], Juat Chin Foo[3], Hong Yu Li[4], Soumita Ghosh[5], Heike Wollmann[1], Amaury Cazenave-Gassiot[2,3], Chee Bing Ong[6], Markus R Wenk[2,3], Weiping Han[1,2,4], Hyungwon Choi[5], Philipp Kaldis[1,2,7]*

[1]Institute of Molecular and Cell Biology (IMCB), A*STAR (Agency for Science, Technology and Research), Singapore, Singapore; [2]Department of Biochemistry, Yong Loo Lin School of Medicine, National University of Singapore (NUS), Singapore, Singapore; [3]Singapore Lipidomics Incubator (SLING), Life Sciences Institute, National University of Singapore (NUS), Singapore, Singapore; [4]Laboratory of Metabolic Medicine, Singapore Bioimaging Consortium (SBIC), A*STAR, Singapore, Singapore; [5]Department of Medicine, Yong Loo Lin School of Medicine, National University of Singapore (NUS), Singapore, Singapore; [6]Biological Resource Centre (BRC), A*STAR, Singapore, Singapore; [7]Department of Clinical Sciences, Lund University, Clinical Research Centre (CRC), Malmö, Sweden

*For correspondence:
philipp.kaldis@med.lu.se

Competing interests: The authors declare that no competing interests exist.

**Abstract** Cell cycle progression and lipid metabolism are well-coordinated processes required for proper cell proliferation. In liver diseases that arise from dysregulated lipid metabolism, hepatocyte proliferation is diminished. To study the outcome of CDK1 loss and blocked hepatocyte proliferation on lipid metabolism and the consequent impact on whole-body physiology, we performed lipidomics, metabolomics, and RNA-seq analyses on a mouse model. We observed reduced triacylglycerides in liver of young mice, caused by oxidative stress that activated FOXO1 to promote the expression of *Pnpla2*/ATGL. Additionally, we discovered that hepatocytes displayed malfunctioning β-oxidation, reflected by increased acylcarnitines (ACs) and reduced β-hydroxybutyrate. This led to elevated plasma free fatty acids (FFAs), which were transported to the adipose tissue for storage and triggered greater insulin secretion. Upon aging, chronic hyperinsulinemia resulted in insulin resistance and hepatic steatosis through activation of LXR. Here, we demonstrate that loss of hepatocyte proliferation is not only an outcome but also possibly a causative factor for liver pathology.

## Introduction

Lipid metabolism is closely linked to cell proliferation, especially since cell division requires the synthesis of phospholipids (PLs) that make up the plasma membrane. Earlier studies highlighted the requirement for synthesis of phosphatidylcholine, a major PL of the cell membrane, in cell cycle progression (*Tercé et al., 1994*). More recently, it was discovered that de novo fatty acid synthesis was essential to provide the fatty acids needed for PL synthesis because inhibition of fatty acid synthesis led to cell cycle arrest at the $G_2$/M transition and prevented the exit from mitosis (*Scaglia et al., 2014*). This was supported by the finding that enzymes involved in fatty acid synthesis were more thermally stable in mitosis and early $G_1$ phase (*Becher et al., 2018*). Therefore, it is not surprising

that a close coordination between lipid metabolism and cell cycle progression is required. Nevertheless, how this coordination is ensured at a molecular level remains to be determined.

The liver, as the metabolic center of the body, is constantly exposed to toxins that can trigger parenchymal hepatocyte cell death. To replace the dying cells, fully differentiated hepatocytes self-renew to regenerate the liver (*Miyaoka and Miyajima, 2013*), with up to 0.5% of hepatocytes dividing at any particular time in non-diseased liver (*Macdonald, 1961*). Nevertheless, it is known that hepatocytes exhibit proliferative defects in the diseased liver (*Yang et al., 2001*; *Zhao et al., 2002*; *Veteläinen et al., 2007*). For example, in patients with non-alcoholic steatohepatitis (NASH), oxidative stress activates the DNA damage checkpoint resulting in cell cycle arrest (*Gentric et al., 2015*). Hepatic steatosis can also cause premature replicative senescence by promoting chronic liver damage and inducing hepatocytes to cycle continuously until they senesce (*Han et al., 2008*), with liver samples from patients diagnosed with non-alcoholic fatty liver disease (NAFLD) or cirrhosis exhibiting increased senescence markers and reduced telomere length (*Kitada et al., 1995*; *Wiemann et al., 2002*; *Aravinthan et al., 2013a*). While this could imply that the loss of hepatocyte proliferation occurs in the liver as a result of liver disease, looking at it from a different angle, the association between senescence and liver disease can also be indicative that a block of hepatocyte proliferation could exacerbate liver disease. Intriguingly, it has been shown that the induction of senescence in hepatocytes causes age-dependent hepatic steatosis through an impairment of fatty acid oxidation (FAO; *Ogrodnik et al., 2017*). However, little else is known about how blocking of the cell cycle affects lipid metabolism and the ensuing impact on whole body metabolism.

p21$^{Cip1/Waf1}$ has been observed to be overexpressed consistently in several types of liver disease (*Aravinthan et al., 2013a*; *Aravinthan et al., 2013b*) and cholangiopathies (*Ferreira-Gonzalez et al., 2018*) and inhibits the activity of cyclin-dependent kinases (CDKs). CDK1 is a cell cycle regulator essential for mitosis (*Lohka et al., 1988*) with inhibition of CDK1 activity resulting in cell cycle arrest, blocked cell proliferation, and senescence (*Itzhaki et al., 1997*; *Diril et al., 2012*). We have previously shown that by specifically deleting *Cdk1* in hepatocytes using Albumin-Cre (*Cdk1* cKO), we were able to prevent hepatocytes from undergoing cell division in vivo (*Diril et al., 2012*). Consequently, these hepatocytes underwent hypertrophy during regeneration after partial hepatectomy, leading to remodeling of glucose metabolism (*Caldez et al., 2018*).

Here, we use the same mouse model of impaired hepatocyte proliferation, aiming to delineate the impact of loss of proliferation on lipid metabolism in the liver and how this affects whole body physiology. Through the use of lipidomics, metabolomics, RNA-seq, ChIP, and biochemical assays, we discovered that *Cdk1* cKO liver contains less triacylglycerides (TGs) as a result of oxidative stress-mediated lipolysis. Furthermore, arrested hepatocytes exhibit defective FAO, leading to release of FFAs into the blood that get stored as TGs in the adipose tissue and maybe in other tissues. The increase in FFAs in the blood triggers chronic hyperinsulinemia which, over time, results in insulin resistance, hepatic steatosis, and the progression of liver disease. Thus, we present evidence supporting the idea that liver disease is not only the cause for impairment of hepatocyte proliferation but it may also be an outcome.

## Results

### Oxidative stress mediates lipolysis in the liver

Since hepatocyte proliferation is known to contribute to the maintenance and regeneration of the liver and we had previously shown that loss of hepatic CDK1 leads to changes in lipid metabolism upon induction of liver regeneration (*Miettinen et al., 2014*), we performed comprehensive lipidomic analyses using mass spectrometry on liver from 8-week-old Alb-Cre, which we henceforth denote as control (Ctrl), and *Cdk1* cKO mice. After QC filtering, we reliably measured 237 lipid species (TGs, PLs, and ACs). PLs, which comprised the majority of the measured species (196 of the 237 species), were generally unchanged (*Supplementary file 1*). The most apparent change in lipid profile from the lipidomics data was that *Cdk1* cKO liver contained less TGs than control liver (*Figure 1A*, *Supplementary file 1*), which we separately confirmed with biochemical TG assays (*Figure 1B*; p=0.0022). This was despite both control and *Cdk1* cKO mice eating the same amount of food (*Figure 1C*; p=0.6825), implying that this change was not due to a difference in food intake. We then checked for TG levels in hepatocytes isolated from 8-week-old mice and noted that *Cdk1*

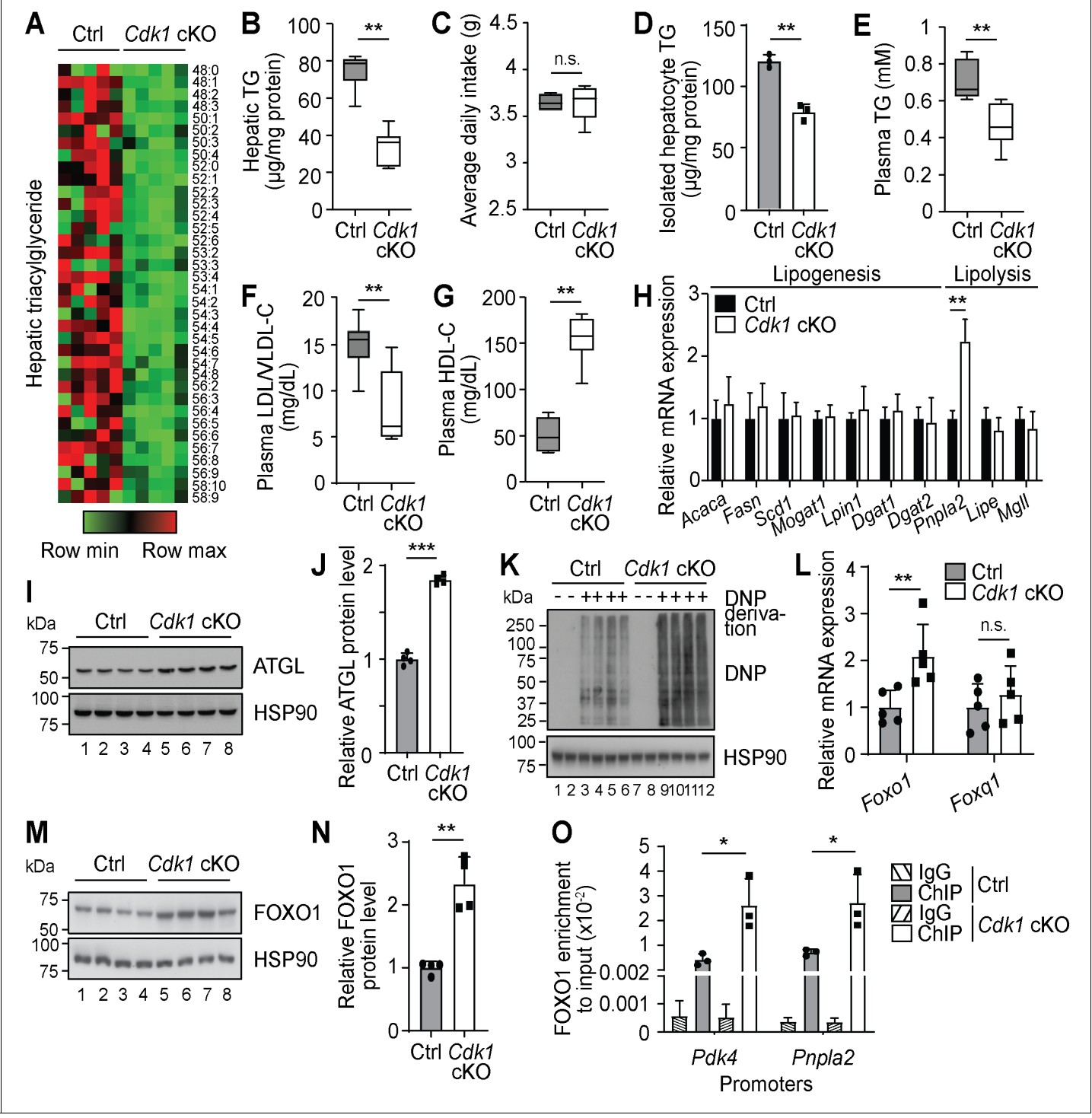

**Figure 1.** Oxidative stress results in Foxo1-dependent activation of *Pnpla2* in young *Cdk1* cKO liver. (**A**) Heat map of triacylglyceride (TG) species in whole liver of control (Ctrl) and *Cdk1* cKO mice as measured by mass spectrometry (lipidomics). Data is available in *Supplementary file 1*. TG species are also ordered as in *Supplementary file 1*. (**B**) Hepatic TG levels as quantified by biochemical assays (n = 6 per genotype). (**C**) Food intake of control and *Cdk1* cKO mice tracked over 2 weeks (6- to 8-week-old) and represented as average daily intake. (**D**) TG from isolated primary hepatocytes (n = 3 per genotype), (**E**) plasma TG, (**F**) plasma LDL/VLDL-cholesterol and (**G**) plasma HDL-cholesterol levels as quantified by biochemical assays (n = 6 per genotype for plasma biochemical assays). (**H**) qPCR for lipogenic and lipolytic genes in whole liver (n = 6 per genotype). (**I**) Immunoblot of liver lysates for ATGL, with HSP90 as loading control. (**J**) Quantification of ATGL protein levels from immunoblot. Error bars represent S.D. (**K**) Immunoblot of liver lysates for carbonylated proteins, probed for Dinitrophenol (DNP) hydrazone after DNP derivation, with HSP90 as loading control. (**L**) qPCR for *Foxo1*

*Figure 1 continued on next page*

Figure 1 continued

and *Foxq1* in whole liver (n = 5 per genotype). (**M**) Immunoblot of liver lysates for FOXO1, with HSP90 as loading control. (**N**) Quantification of FOXO1 protein levels from immunoblot. Error bars represent S.D. (**O**) Enrichment of FOXO1 localization at promoters of *Pdk4* and *Pnpla2* in isolated primary hepatocytes upon ChIP-qPCR after normalization to input (n = 3 per genotype). Error bars represent S.E.M. unless otherwise indicated. All experiments were performed on tissue samples or cells from 8-week-old mice. All source data are available in *Supplementary file 10* unless specifically indicated. The online version of this article includes the following figure supplement(s) for figure 1:

**Figure supplement 1.** NAC treatment and silencing of *Pnpla2* can reverse the reduction of triacylglyceride (TG) phenotype.

cKO hepatocyte TG levels were significantly decreased as well (*Figure 1D*; p=0.0012), illustrating that the reduction in whole liver TG levels were, in part, due to the decrease of TGs in hepatocytes. Plasma TG (*Figure 1E*; p=0.0043) and plasma low-density lipoproteins (LDL)/very-low-density lipo-proteins (VLDL) were also reduced (*Figure 1F*; p=0.0043) with a corresponding increase in high-den-sity lipoproteins (*Figure 1G*; p=0.0022) in *Cdk1* cKO liver compared to control. As VLDLs are the main mode of hepatic TG secretion into the blood stream (*Choi and Ginsberg, 2011*), this sug-gested that the reduction in hepatic TGs was not due to an increase in secretion of TGs, but likely a change in TG synthesis or TG breakdown in the liver. To address this, we screened a panel of lipo-genic and lipolytic genes for changes in gene expression, and interestingly, a specific increase of the lipolytic gene *Pnpla2* became evident (*Figure 1H*; p=0.0022). *Pnpla2* encodes the ATGL protein, which is the main rate-limiting enzyme responsible for breaking down TGs (*Zimmermann et al., 2004*), and accordingly, the protein levels of ATGL were increased in *Cdk1* cKO liver (*Figure 1I–J*; p<0.0001). Over-expression of ATGL is sufficient to induce lipolysis in the liver (*Reid et al., 2008*), indicating that the reduced levels of hepatic TGs in *Cdk1* cKO liver might be due to increased lipoly-sis mediated by higher levels of *Pnpla2*/ATGL.

The specific increase of *Pnpla2* but not any other lipolytic genes implies that the upregulation of *Pnpla2* was not a result of a general increase in the lipolytic machinery. Interestingly, *Pnpla2* can be induced by the transcription factor FOXO1 (*Chakrabarti and Kandror, 2009*; *Zhang et al., 2016*), which is activated by mitochondria-dependent oxidative stress (*Lettieri Barbato et al., 2014*). Since *Cdk1* cKO hepatocytes display impaired mitochondrial functions (*Caldez et al., 2018*), we hypothe-sized that the induction of *Pnpla2* might be due to oxidative stress-dependent activation of FOXO1. Indeed, *Cdk1* cKO liver exhibited greater levels of oxidative stress, as seen by the induction of anti-oxidant enzymes *Gpx1*, *Gpx2*, *Gpx3*, and *G6pdx* (*Figure 1—figure supplement 1A*) and the increase in protein carbonylation, which is a good marker of oxidative stress (*Dalle-Donne et al., 2003*; *Figure 1K*). Correspondingly, there was an increase of Foxo1 at both the mRNA (*Figure 1L*; p=0.0079) and protein level (*Figure 1M–N*; p=0.0065), leading to enhanced localization of FOXO1 by chromatin immunoprecipitation (ChIP) at the promoters of *Pnpla2* and *Pdk4*, a well-established target of FOXO1 (*Kwon et al., 2004*) (*Figure 1O*; p=0.0419 for *Pnpla2* promoter and p=0.0275 for *Pdk4* promoter). Because FOXO1 transcriptional activity can be repressed by interaction with FOXQ1 (*Cui et al., 2016*), we determined *Foxq1* expression and discovered that there was no signif-icant difference in the mRNA levels of *Foxq1* (*Figure 1L*; p=0.5476). When we fed *Cdk1* cKO mice with the antioxidant N-acetylcysteine (NAC) to reduce systemic oxidative stress, we observed a reversal in the amount of protein carbonylation (*Figure 1—figure supplement 1B*) and the mRNA expression of antioxidant enzymes (*Figure 1—figure supplement 1C*), *Foxo1*, and *Pnpla2* (*Fig-ure 1—figure supplement 1D*). Furthermore, hepatic TG levels of NAC-fed *Cdk1* cKO mice recov-ered to control levels (*Figure 1—figure supplement 1E*). This data supports the hypothesis that the reduction in hepatic TG levels in *Cdk1* cKO liver was in part due to oxidative stress and this was transmitted by the FOXO1-*Pnpla2*/ATGL axis.

To more conclusively show that the decrease in liver TGs is mediated by ATGL, we attempted to knockdown *Pnpla2* in the liver of *Cdk1* cKO mice. We performed hydrodynamic tail vein injections (*Yokoo et al., 2016*) of a plasmid expressing short hairpin RNA (shRNA) targeting murine *Pnpla2* (pLKO-shPnpla2) to deliver the plasmid to the liver (see Materials and methods). We verified using qPCR that there was an increase of *Pnpla2* in the liver of *Cdk1* cKO mice (*Figure 1—figure supple-ment 1G*), confirming our earlier findings (*Figure 1H*, *Figure 1—figure supplement 1D*). *Cdk1* cKO mice that were injected with pLKO-shPnpla2 displayed a reduction of *Pnpla2* expression in the liver, indicating successful knockdown of *Pnpla2*. We then analysed the hepatic TGs in these mice, and

found that in *Cdk1* cKO mice with *Pnpla2* knockdown, hepatic TG levels were rescued to levels comparable with control (*Figure 1—figure supplement 1H*). In addition, correlation analysis between relative *Pnpla2* mRNA expression and hepatic TG levels (*Figure 1—figure supplement 1I*) proposed a negative linear relationship with a Pearson's correlation coefficient of −0.84 (p-value<0.0001), supporting the well-established TG lipase activity of ATGL (*Zimmermann et al., 2004*; *Reid et al., 2008*) and indicating that the increase of *Pnpla2* in the liver of *Cdk1* cKO mice is likely causative for the observed reduction of TGs.

## Defective FAO in hepatocytes

Another prominent observation from our extensive lipidomics data of the *Cdk1* cKO liver is that all the detected AC species were significantly increased (*Figure 2A*, *Supplementary file 1*). This is mirrored by our lipidomic data on hepatocytes isolated from 8-week-old control and *Cdk1* cKO mice (*Supplementary file 2*), which, in all, reliably measured 240 lipid species. Although not statistically significant for most of the species due to the small sample size (n = 3) and large variability, a trend of increase in AC levels in isolated hepatocytes was observed (*Figure 2B*). AC is the form by which long-chain fatty acids are transported across the mitochondrial membrane into mitochondria for degradation via β-oxidation (*Houten et al., 2016*). The accumulation of ACs in *Cdk1* cKO liver resembles that seen in patients and mouse models with defective mitochondrial FAO (*Houten et al., 2016*; *Lee et al., 2016*), implying FAO might be deficient in *Cdk1* cKO hepatocytes. This was further supported by a decrease in β-hydroxybutyrate (*Figure 2C*; p=0.0002), a metabolic product of β-oxidation (*Grabacka et al., 2016*). When we performed FAO assays on primary hepatocytes from control and *Cdk1* cKO mice, we were able to confirm that *Cdk1* cKO hepatocytes display reduced basal and maximal capacity for performing FAO (*Figure 2D–E*; p<0.0001 for basal FAO capacity and p=0.0018 for maximal FAO capacity).

To explore the cause for this, we measured transcript levels of genes involved in mitochondrial β-oxidation. Unexpectedly, none of the genes tested showed any difference between control and *Cdk1* cKO hepatocytes, including *Ppara* (*Figure 2F*; p=0.1905), the master regulator of FAO (*Grabacka et al., 2016*). Interestingly though, immunoblotting for HADHA and ACADVL, two key enzymes in the FAO pathway, revealed a decrease in protein levels of both enzymes (*Figure 2G–H*; p=0.0286 for both HADHA and ACADVL) despite unchanged transcript levels (*Figure 2F*; p=0.5476 for *Acadvl* and p=0.8414 for *Hadha*). This might be potentially explained by CDK1 modulation of mitochondrial import via phosphorylation of the outer membrane protein TOM6 (*Harbauer et al., 2014*). In the absence of CDK1, reduced mitochondrial import of FAO enzymes could result in cytoplasmic accumulation that triggers ubiquitin-dependent degradation of these enzymes (*Habelhah et al., 2004*; *Bragoszewski et al., 2013*). A more recent finding showed that CDK1 can also enhance fatty acid import into the mitochondria by phosphorylating SIRT3 and augmenting SIRT3-mediated CPT2 dimerization into the functional fatty acid import complex to promote FAO (*Liu et al., 2020*). We checked whether this mechanism was relevant in our system. Immunoprecipitation of SIRT3 followed by immunoblotting for phosphorylated Ser/Thr affirmed that SIRT3 was hypophosphorylated in *Cdk1* cKO liver (*Figure 2I*) with no impact on total SIRT3 protein levels (*Figure 2J*). In addition, we detected increased acetylated residues upon immunoprecipitation of CPT2 and probing for pan-acetylation (*Figure 2K*). Taken together, we discovered that *Cdk1* cKO hepatocytes display decreased FAO, possibly due to reduced mitochondrial import of FAO enzymes as well as fatty acids that are the substrates of these enzymes.

## Liver FFAs are released into the blood stream and affect peripheral tissues

With a continuous increase of non-esterified FFAs due to persistent lipolysis (*Figure 1*) and a block in fatty acid degradation (*Figure 2*) in the liver, we investigated whether there was an increase in secretion of FFAs into the blood stream. Mass spectrometry analysis of plasma samples showed an increase in many detected FFA species in the plasma of *Cdk1* cKO mice (*Figure 3A*, *Supplementary file 3*). Notably, treatment of *Cdk1* cKO mice with NAC, which reversed the hepatic TG phenotype (*Figure 1—figure supplement 1E*), did not revert plasma FFA back to control levels (*Figure 1—figure supplement 1F*), although levels were insignificantly yet visibly lower. This would suggest that the increase in plasma FFAs is in part dependent on the lipolysis of TGs mediated by

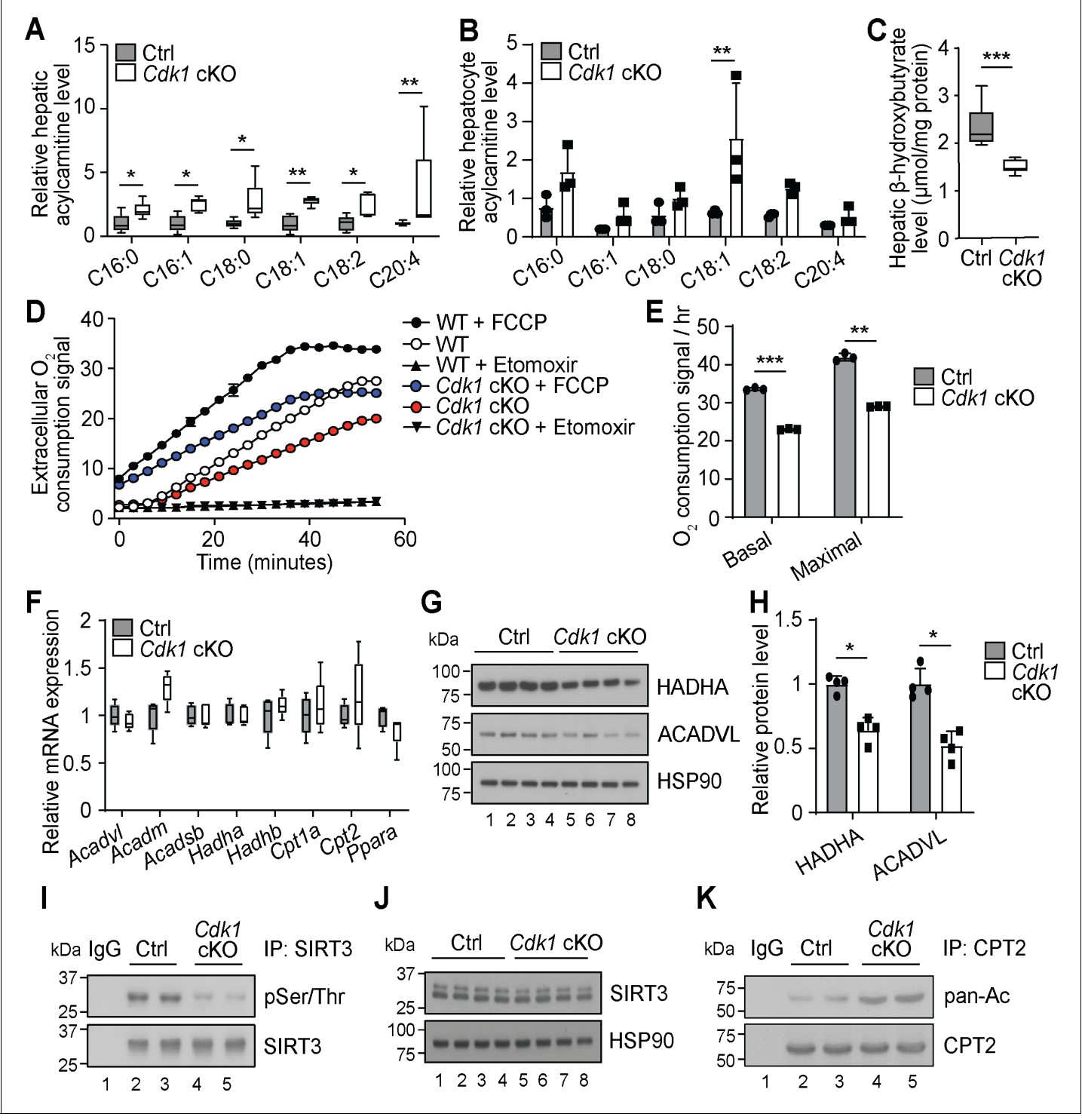

**Figure 2.** *Cdk1* cKO hepatocytes have reduced capacity for fatty acid oxidation (FAO). Relative acylcarnitine levels in liver (A; n = 5 per genotype) and primary hepatocytes (B; n = 3 per genotype) of control (Ctrl) and *Cdk1* cKO mice as measured by mass spectrometry. Data is available in *Supplementary file 1* and *Supplementary file 2*, respectively. (C) Hepatic β-hydroxybutyrate levels measured using biochemical assays (n = 8 per genotype). Error bars represent S.E.M. (D) FAO assays on isolated primary hepatocytes that were untreated, treated with FCCP to maximize oxidative capacity or treated with Etomoxir to block FAO. (E) Quantification of basal and maximal FAO capacity of isolated hepatocytes based on oxygen consumption signal of untreated and FCCP-treated hepatocytes, respectively, from the FAO assays. Statistical significance was calculated using unpaired two-tailed t-test with Welch's correction. (F) qPCR for expression of FAO genes in whole liver (at least n = 4 per genotype). Error bars represent S.E.M. (G) Immunoblot of liver lysate, probed for HADHA and ACADVL with HSP90 as loading control. (H) Quantification of HADHA and

*Figure 2 continued on next page*

*Figure 2 continued*

ACADVL protein levels from immunoblot. (I) SIRT3 was immunoprecipitated from lysates of Ctrl or *Cdk1* cKO liver and probed for phosphorylated Ser/ Thr (pSer/Thr). IgG was used as negative control. (J) Immunoblot of liver lysate, probed for SIRT3 with HSP90 as loading control. (K) CPT2 was immunoprecipitated from lysates of Ctrl or *Cdk1* cKO liver and probed for pan-acetylated residues (pan-Ac). IgG was used as negative control. Error bars represent S.D. unless otherwise indicated. All experiments were performed on tissue samples or cells from 8-week-old mice. All source data are available in *Supplementary file 10* unless specifically indicated.

ATGL, and in part also dependent on impaired FAO in *Cdk1* cKO hepatocytes. As plasma FFA can be taken up and stored as TGs in white adipose tissue (WAT), we investigated whether the increase in plasma FFA resulted in an increase in storage of FFA in the adipose tissue. MRI revealed an overall increase in fat mass in *Cdk1* cKO mice (*Figure 3B*; p=0.0079), with a higher subcutaneous white adipose tissue (scWAT) weight to body weight ratio and epididymal white adipose tissue (epWAT) weight to body weight ratio (*Figure 3C*; p<0.0001 for both scWAT and epWAT to body weight ratio). This might, in part, explain the more pronounced increase in body weight of *Cdk1* cKO mice (*Figure 3D–E*) despite the similar food intake between both groups of mice (*Figure 1C*). Histological staining of the scWAT and epWAT (*Figure 3F*) indicated that adipocytes were significantly increased in size, from 494 (±49)μm$^2$ in control scWAT to 692 (±63)μm$^2$ in *Cdk1* cKO scWAT and 589 (±86)μm$^2$ in control epWAT to 1066 (±153)μm$^2$ in *Cdk1* cKO epWAT (*Figure 3G*; p=0.0079 for both scWAT and epWAT). Biochemical assays further confirmed that *Cdk1* cKO WAT contained higher levels of TGs (*Figure 3H*; p=0.0079 for both scWAT and epWAT). When we then probed *Cdk1* cKO WAT for the expression of genes involved with fatty acid synthesis, we found that *Acaca*, *Fasn*, and *Scd1* were decreased (*Figure 3I–J*), implying that the increase in TGs was not due to increases in WAT fatty acid synthesis. Instead, this repression of de novo lipogenesis parallels what is seen in mice fed with high fat diet, whereby there is elevated supply of dietary fat to the WAT via the blood stream (*Shillabeer et al., 1990*; *Tovar et al., 2011*). These results thus suggest that the increase of plasma FFA led to elevated storage of FFA as TGs in WAT of *Cdk1* cKO mice.

Higher levels of plasma FFA can affect peripheral tissue besides WAT. For example, the expression of FAO enzymes in the skeletal muscle are boosted in the presence of elevated plasma FFA levels (*Garcia-Roves et al., 2007*). In accordance with this, we observed increased expression of various FAO-associated genes in the skeletal muscle of *Cdk1* cKO mice compared to control mice (*Figure 4A*). Of particular interest is the fact that plasma FFAs are also known to promote insulin secretion by pancreatic β-cells even in the fasting state (*Itoh et al., 2003*; *Cen et al., 2016*). Interestingly, we detected increased plasma insulin in *Cdk1* cKO mice (*Figure 4B*; p=0.0499), a state known as hyperinsulinemia, despite there being no difference in the transcript level of *Ins2* between *Cdk1* cKO and control pancreas (*Figure 4C*; p=0.8413). Furthermore, *Cdk1* cKO mice displayed reduced blood glucose levels (*Figure 4D*; p<0.0001) and greater hepatic glycogen content (*Figure 4E*; p=0.0079), both of which are phenotypes associated with enhanced insulin signaling. This was confirmed by immunoblotting (*Figure 4F*), where we observed enhanced phosphorylation of INSRB (*Figure 4G*; p=0.0312) and AKT (*Figure 4H*; p=0.0042) in *Cdk1* cKO liver. When we performed glucose tolerance tests (*Figure 4I*), *Cdk1* cKO mice were more glucose tolerant and responded better to exogenous glucose, as seen from a lower area under the curve (*Figure 4J*; p<0.0001), which is likely a result of the increased insulin levels (*Figure 4B*). Taken together, our data suggests that the elevated plasma FFA promotes insulin secretion and maintains a hyperinsulinemic condition in *Cdk1* cKO mice.

## Aged mice exhibit insulin resistance and liver disease

Chronic hyperinsulinemia can lead to insulin resistance in patients (*Morita et al., 2017*), hence we wondered whether plasma insulin remained high in *Cdk1* cKO mice over time. We analyzed plasma insulin levels of 6- and 12-month-old mice and found that plasma insulin was increased from an average of 0.6 ng/mL in control mice to 1.4 ng/mL in *Cdk1* cKO mice at 6 months old and from an average of 1 ng/mL in control mice to 2 ng/mL in *Cdk1* cKO mice at 12 months old (*Figure 5A*; p=0.0079 for both 6 and 12 months). This paralleled the consistently higher levels of plasma FFAs in *Cdk1* cKO mice compared to age-matched control mice at 6 and 12 months (*Figure 5B*; p=0.0079 for both 6 and 12 months). When we checked for insulin signaling in the aged liver by immunoblotting (*Figure 5C*), we noticed the levels of total INSRB protein was reduced (*Figure 5D*; p=0.0220),

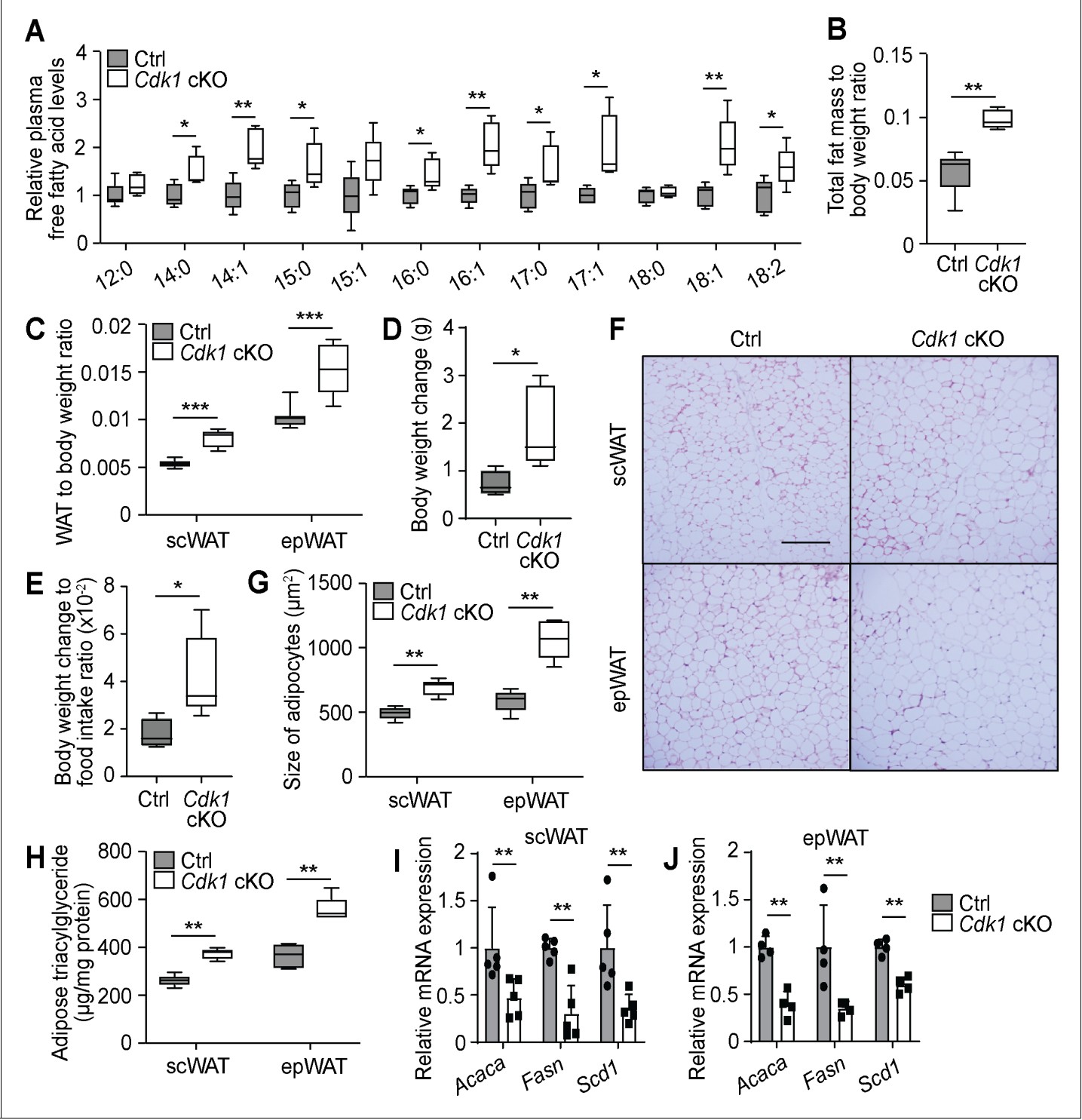

**Figure 3.** Free fatty acids (FFAs) are transported to the adipose tissue for storage in young *Cdk1* cKO mice. (A) Relative plasma FFA levels in control (Ctrl) and *Cdk1* cKO mice as measured by mass spectrometry (n = 5 per genotype). Data is available in ***Supplementary file 3***. (B) Ratio of total fat mass, as measured by magnetic resonance imaging, to body weight (n = 5 for each genotype). (C) Ratio of subcutaneous WAT (scWAT) or epididymal WAT (epWAT) weight to body weight (n = 10 per genotype). (D) Body weight change between start and end of 2-week food intake tracking period. (E) Ratio of body weight change to food intake. (F) Representative H and E image of scWAT and epWAT from control and *Cdk1* cKO mice. Scale bar represents 100 µm in all panels. (G) Quantification of size of adipocytes from H and E images (at least 500 adipocytes per mouse were measured, n = 5 per genotype). (H) Adipose triacylglyceride levels from scWAT and epWAT measured using biochemical assays (n = 5 per genotype). qPCR for lipogenic genes in scWAT (I) and epWAT (J) of control and *Cdk1* cKO mice (at least n = 4 per genotype). Error bars for all graphs represent S.E.M. All

*Figure 3 continued on next page*

*Figure 3 continued*

experiments were performed on tissue samples from 8-week-old mice. All source data are available in *Supplementary file 10* unless specifically indicated.

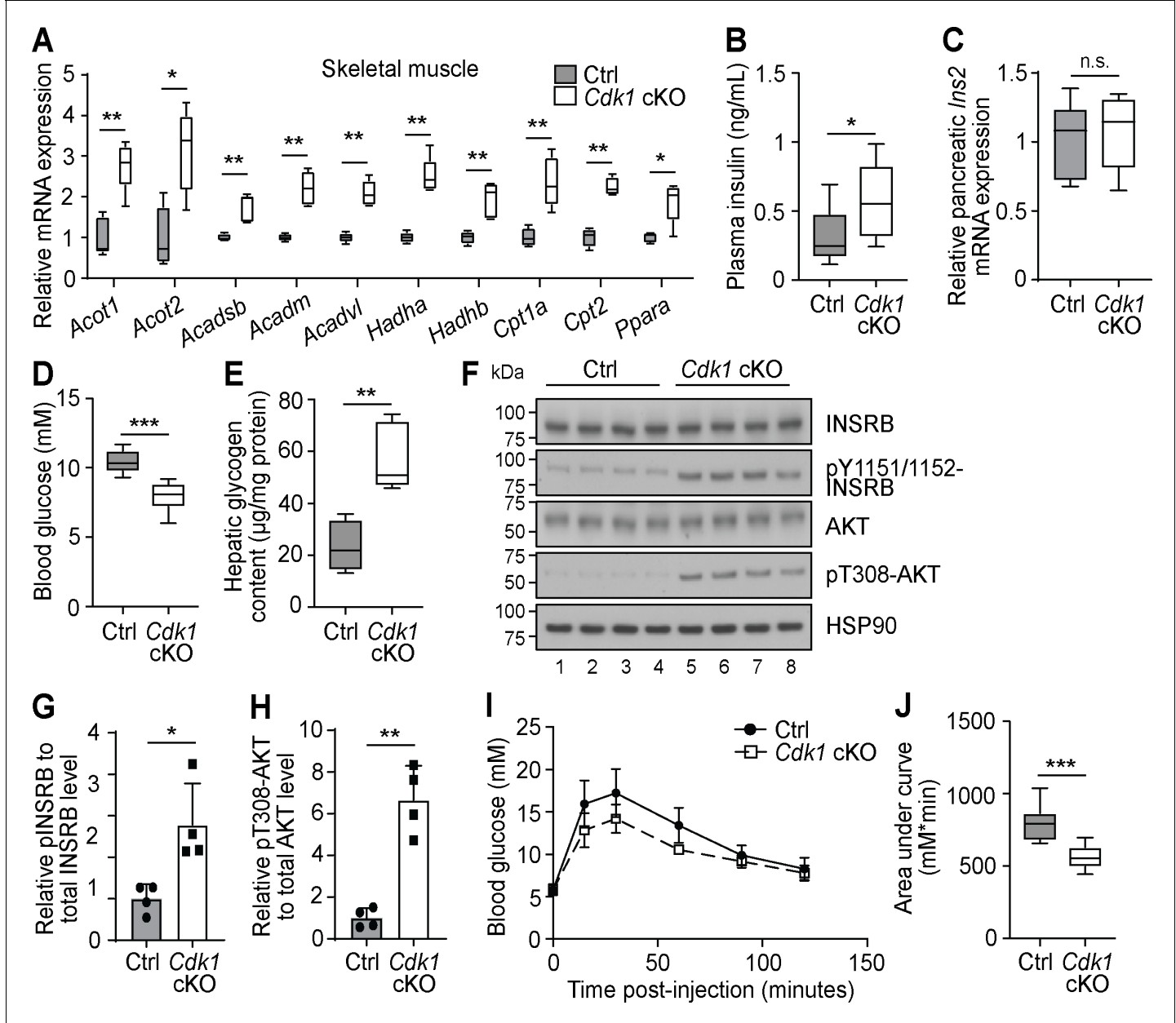

**Figure 4.** Increased plasma fatty acids affect peripheral tissues and induce insulin signaling in young *Cdk1* cKO mice. (**A**) qPCR for fatty acid oxidation genes in skeletal muscle (quadriceps) from control (Ctrl) and *Cdk1* cKO mice (n = 5 per genotype). (**B**) Plasma insulin level examined with ELISA (n = 8 per genotype). (**C**) qPCR for *Ins2* in pancreas of Ctrl and *Cdk1* cKO mice (n = 5 per genotype). (**D**) Fed state blood glucose measurements using glucose meter (at least n = 8 per genotype). (**E**) Hepatic glycogen content measured with biochemical assays (n = 5 per genotype). (**F**) Immunoblot of liver lysate probed for INSRB, phosphorylated INSRB at Y1151/1152 (pY1151/1152-INSRB), AKT, and phosphorylated AKT at T308 (pT308-AKT). HSP90 serves as loading control. (**G**) Quantification of pY1151/1152-INSRB levels normalized to total INSRB protein and (**H**) pT308-AKT levels normalized to total AKT protein from immunoblot. Error bars for immunoblot quantifications represent S.D. Blood glucose measurements (**I**) and area under curve analysis (**J**) from intraperitoneal glucose tolerance test (at least n = 10 per genotype). Error bars for all graphs represent S.E.M. unless otherwise indicated. All experiments were performed on tissue samples from 8-week-old mice. All source data are available in *Supplementary file 10*.

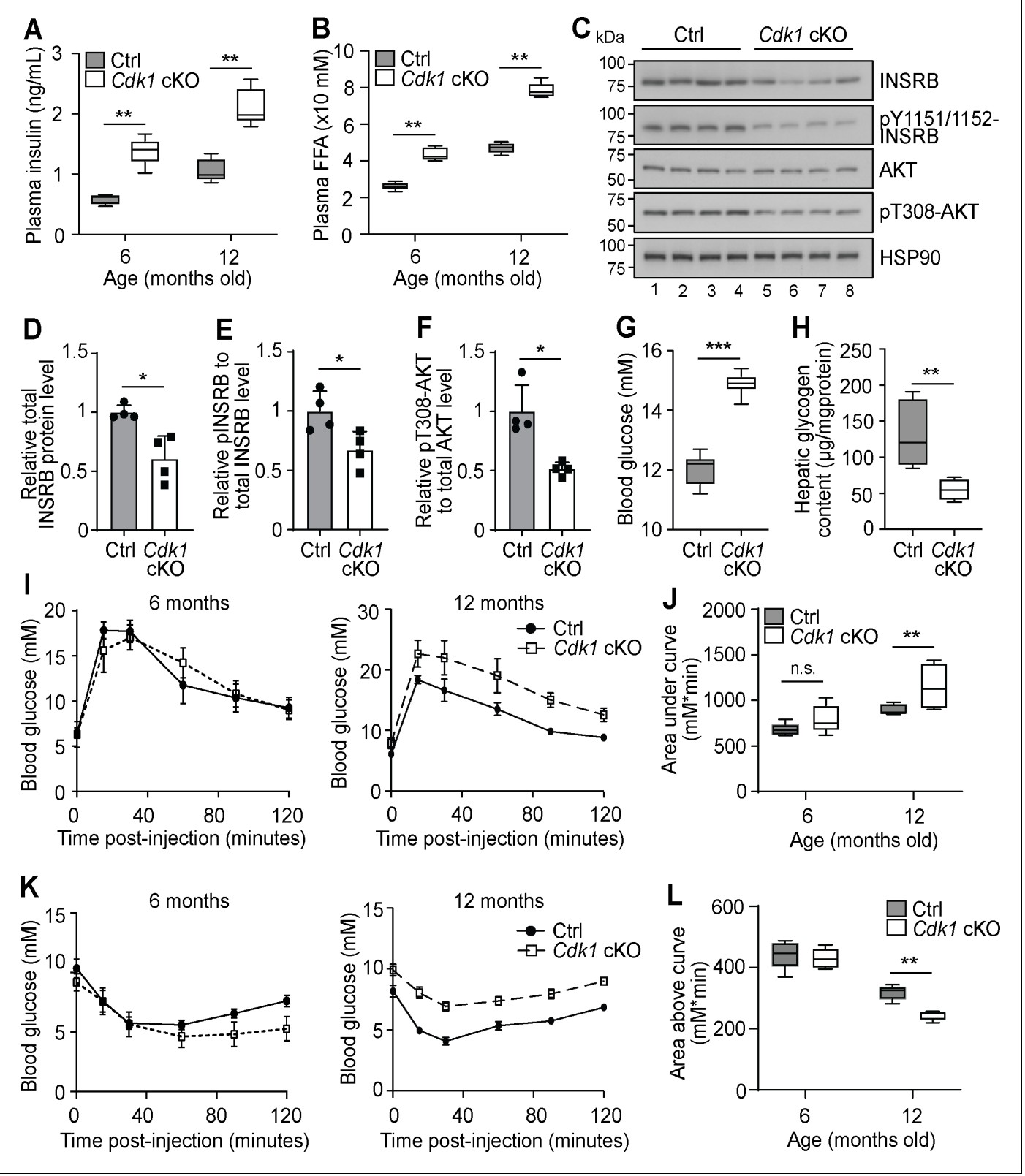

**Figure 5.** Aged *Cdk1* cKO mice develop insulin resistance. (**A**) Plasma insulin levels measured at 6 and 12 months old by ELISA (n = 5 per genotype per age). Error bars represent S.E.M. (**B**) Plasma FFA levels measured at 6 and 12 months old by biochemical assays (n = 5 per genotype per age). Error bars represent S.E.M. (**C**) Immunoblot of liver lysates from 12-month-old mice, probed for INSRB, phosphorylated INSRB at Y1151/1152 (pY1151/1152-INSRB), AKT, and phosphorylated AKT at T308 (pT308-AKT), with HSP90 as loading control. Quantification of total INSRB protein levels (**D**), pY1151/

*Figure 5 continued on next page*

*Figure 5 continued*

1152-INSRB levels normalized to total INSRB protein (E), and pT308-AKT normalized to total AKT protein (F) from immunoblot. (G) Fed state blood glucose measurements (at least n = 8 per genotype) and (H) hepatic glycogen content of 12-month-old mice (n = 5 per genotype). Error bars represent S.E.M. (I) Blood glucose measurements and (J) area under curve analysis from intraperitoneal glucose tolerance test (ipGTT) of 6- and 12-month-old mice (at least n = 5 per genotype). (K) Blood glucose measurements and (L) area above curve analysis from intraperitoneal insulin tolerance test (ipITT) of 6- and 12-month-old mice (n = 5 per genotype). Error bars represent S.D. unless otherwise stated. All source data are available in *Supplementary file 10*.

The online version of this article includes the following figure supplement(s) for figure 5:

**Figure supplement 1.** Adipose tissue from aged *Cdk1* cKO mice exhibit reduced insulin signaling.

which might be due to chronic exposure to hyperinsulinemia (*Ronnett et al., 1982*; *Palsgaard et al., 2009*) that can, in part, lead to insulin resistance. We also detected hypophosphorylated INSRB, even after normalizing to total INSRB levels (*Figure 5E*; p=0.0289), and hypophosphorylated AKT (*Figure 5F*; p=0.0191), suggesting that insulin signaling was impaired in the liver despite higher plasma insulin levels. A similar observation was made from immunoblots of epWAT collected from aged *Cdk1* cKO mice (*Figure 5—figure supplement 1A*), whereby we noted decreased INSRB protein (*Figure 5—figure supplement 1B*; p=0.0253) and hypophosphorylated INSRB (*Figure 5—figure supplement 1C*; p<0.0001) and AKT (*Figure 5—figure supplement 1D*; p=0.0034), hinting at defective insulin signaling in adipose tissues as well. This was supported by the change in expression of a number of adipokines in the epWAT. In particular, adiponectin (*Adipoq*) expression was diminished and resistin (*Retn*) expression was increased (*Figure 5—figure supplement 1E*), both of which are associated with insulin resistance (*Kadowaki, 2006*; *Jiang et al., 2016*), especially since adiponectin represses the expression of gluconeogenic genes in hepatocytes and promotes membrane localization of GLUT4 in myocytes to increase the uptake of glucose (*Combs et al., 2001*; *Yamauchi et al., 2002*). We also observed an increase in expression of *Fgf21* and *Gdf15* (*Figure 5—figure supplement 1E*), which reflects the state of metabolic stress that the mice are in *Manoli et al., 2018*; *Patel et al., 2019*. Aged *Cdk1* cKO mice displayed increased blood glucose levels (*Figure 5G*; p<0.0001) and reduced hepatic glycogen content (*Figure 5H*; p=0.0079), indicating that 12-month-old *Cdk1* cKO mice had developed insulin resistance. When we performed glucose and insulin tolerance tests in 12-month-old mice, *Cdk1* cKO mice were more glucose intolerant (*Figure 5I–J*; p=0.0057) and less insulin sensitive (*Figure 5K–L*; p=0.0079) than control mice. Taken together, aged *Cdk1* cKO mice develop insulin resistance, possibly due to chronic hyperinsulinemia.

Hepatic steatosis is present in up to 50% of patients with type 2 diabetes (*Roden, 2006*). Thus, we investigated whether steatosis had developed in the liver of aged *Cdk1* cKO mice. Histopathological analysis of hematoxylin and eosin (H and E)-stained liver sections from 12-month-old mice by a certified pathologist revealed the presence of macrovesicular fatty changes in aged *Cdk1* cKO mice, with control mice scoring 0, indicating the lack of any observable fatty changes, while *Cdk1* cKO mice were more variable with an average score of 1.8, but with some scoring as high as 4 (*Supplementary file 4*). This was confirmed by Oil Red O staining that illustrated the presence of lipid droplets in most aged *Cdk1* cKO mice but not in any of the aged control mice (*Figure 6A*). Additionally, TG assays (*Figure 6B*; p=0.0079) confirmed higher levels of hepatic TG in *Cdk1* cKO liver relative to control. There was also greater expression of *Acaca*, *Fasn*, *Mogat1*, and *Cidec* (*Figure 6C*), which are genes associated with lipogenesis and lipid accumulation, as well as increased protein level of the lipogenic transcription factor SREBP1c (*Figure 6D–E*; p<0.0001). Even at 12 months old, lipidomics analysis (390 species detected) indicated that there were elevated levels of nearly all the 180 TG species detected (*Figure 6F*, *Supplementary file 5*), which could be correlated to the presence of insulin resistance (*Figure 5*). Hence, we present convincing data that aged *Cdk1* cKO mice develop hepatic steatosis, possibly through the upregulation of the lipogenic pathway but the mechanisms for this need to be further investigated (see below).

About 20% of patients with NAFLD progress to NASH, characterized by the development of fibrosis (*Farrell and Larter, 2006*). Therefore, we also probed whether aged *Cdk1* cKO mice develop liver fibrosis. Sirius Red staining indicated that aged *Cdk1* cKO mice exhibited more fibrotic areas compared to control (*Figure 6G*; p=0.0043), with some mice developing parenchymal and bridging fibrosis (*Figure 6A*). In our histopathology analysis, aged control mice scored 0, whereas aged *Cdk1* cKO mice had an average score of 1.4, with some scoring up to 3 (*Supplementary file*

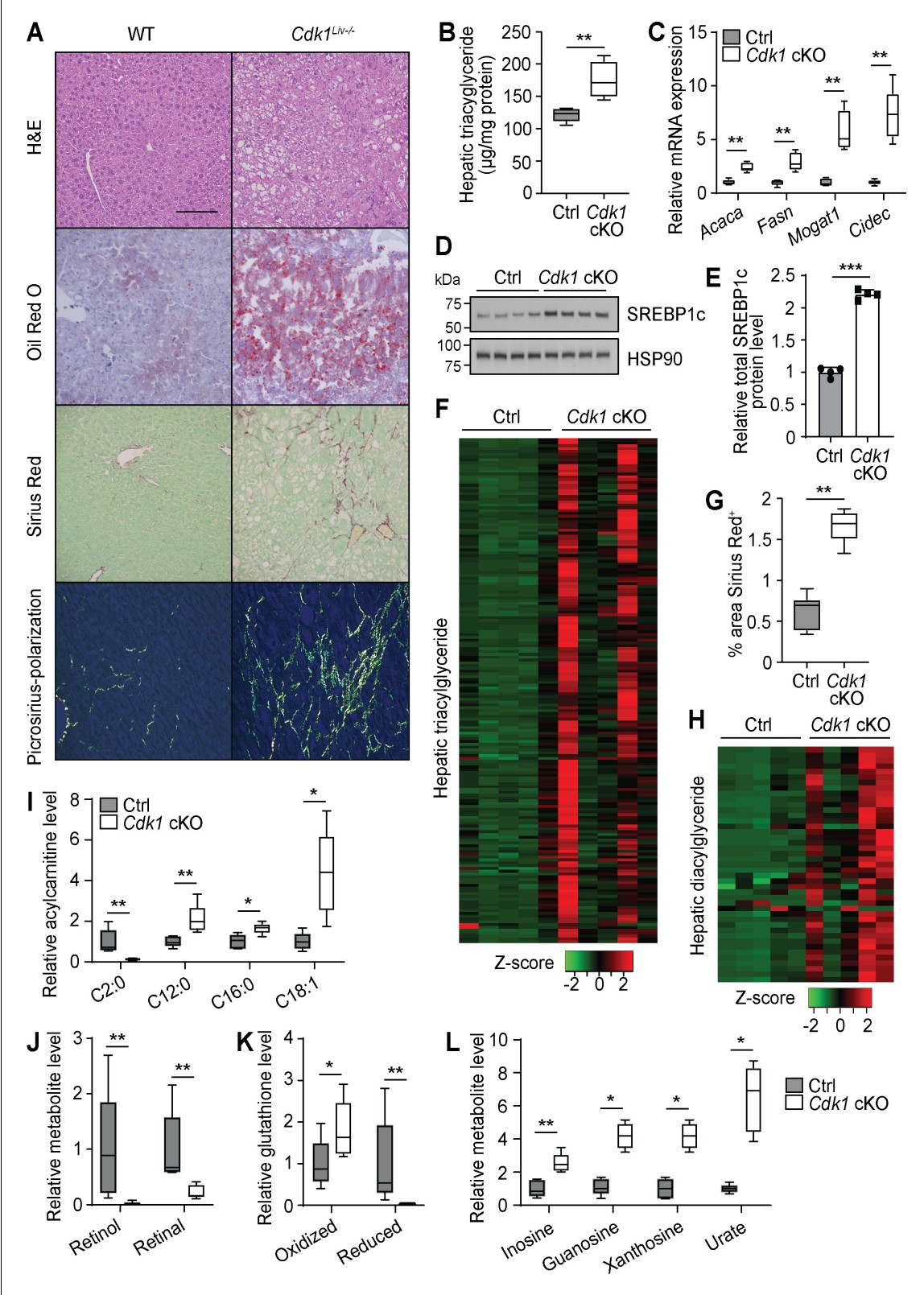

**Figure 6.** Aged *Cdk1* cKO mice develop hepatic steatosis progressing to steatohepatitis. (**A**) Representative H&E, Oil Red O, and Sirius Red staining of 12-month-old control (Ctrl) and *Cdk1* cKO liver sections. Scale bar represents 100 μm in all panels. (**B**) Hepatic triacylglycerides (TGs) measured by biochemical assays (n = 5 per genotype). (**C**) qPCR of genes associated with lipogenesis and lipid accumulation in whole liver (n = 5 per genotype). (**D**) Immunoblot of liver lysates from 12-month-old mice, probed for SREBP1c. HSP90 image is reused from *Figure 5C* as samples were ran on the same

*Figure 6 continued on next page*

*Figure 6 continued*

blot. (E) Quantification of SREBP1c protein level from immunoblot. Error bars represent S.D. (F) Heat map of TG species in whole liver of 12-month-old Ctrl and *Cdk1* cKO mice as measured by mass spectrometry (lipidomics). Data is available in *Supplementary file 5*. (G) Quantification of percentage of Sirius-Red-positive area (five images per mouse, n = 5 per genotype). (H) Heat map of diacylglyceride species in whole liver of Ctrl and *Cdk1* cKO mice as measured by mass spectrometry (metabolomics). Relative levels of acylcarnitines (I), retinol metabolism metabolites (J), glutathione redox status (K), and purine catabolism metabolites (L) from metabolomics data. Metabolomics data was previously published (*Narayanaswamy et al., 2020*). Error bars represent S.E.M. unless otherwise stated. All source data (besides metabolomics data) are available in *Supplementary file 10*.

The online version of this article includes the following figure supplement(s) for figure 6:

**Figure supplement 1.** Purine catabolism pathway.

*4*). Coupled to the presence of hepatocyte ballooning in *Cdk1* cKO liver (*Figure 6A*), another histological feature of NASH (*Lackner, 2011*), our histological data supports the progression to a NASH-like state in aged *Cdk1* cKO liver.

We recently published a detailed metabolomics analysis of aged control and *Cdk1* cKO liver (*Narayanaswamy et al., 2020*) and wanted to perform a more in-depth inspection of this metabolomics data by correlating changes in metabolites to the phenotypes and providing biological context to the findings. There was a substantial amount of lipids identified in the metabolomics data, likely due to the method of metabolite extraction used (*Narayanaswamy et al., 2020*). Nevertheless, in line with our lipidomics data (*Figure 6F*) and histological observations (*Figure 6A*), TGs were increased in *Cdk1* cKO liver, as were diacylglycerides (*Figure 6H*). We also found that most detected AC species were increased in aged *Cdk1* cKO liver (*Figure 6I*), similar to what was seen in 8-week-old *Cdk1* cKO liver (*Figure 2A–B*), which might imply that the loss of FAO in *Cdk1* cKO hepatocytes (*Figure 2C–E*) was sustained in aging. Acetylcarnitine, a subspecies of ACs, was the only AC species reduced and among the most diminished metabolites in aged *Cdk1* cKO liver (*Figure 6I*). This could be due to the diversion of acetyl-CoA toward the lipogenic pathway (*Figure 6C–E*), thereby reducing the need for carnitine to buffer acetyl-CoA levels in cells. Other notable metabolites that were changed in aged *Cdk1* cKO liver compared to control include retinol and retinal (*Figure 6J*), which were the most decreased metabolites observed in the metabolomics data, and both oxidized and reduced forms of glutathione (*Figure 6K*). Decreases of hepatic retinol and retinal (*Figure 6J*) might be an outcome of hepatic stellate cell activation, as evidenced by the increased fibrosis (*Figure 6A*), and the resultant loss of lipid droplets in these cells, which are the main stores of retinoids in the liver (*Narayanaswamy et al., 2020*). The increase of oxidized glutathione and decrease of reduced glutathione could reflect the altered redox status in *Cdk1* cKO liver that was evident even in young mice (*Figure 1*).

Interestingly, metabolites from the purine metabolism pathway, namely inosine, guanosine, xanthosine, and urate, were elevated (*Figure 6L*, *Figure 6—figure supplement 1*). This is particularly intriguing because adenosine deaminase (ADA), the enzyme that catalyzes the conversion of adenosine to inosine, is known to be more active in type 2 diabetic patients (*Kurtul et al., 2004*) and is associated with liver fibrosis in NAFLD (*Jiang et al., 2018*). Increases in ADA activity can lead to an increased flux through the purine metabolic pathway, eventually leading to increased production of urate by xanthine dehydrogenase (XDH) as the final step in the pathway (*Mandal and Mount, 2015*). In fact, the presence of elevated serum urate level is a biomarker and risk factor for fatty liver disease (*Jensen et al., 2018*), and greater XDH activity has been suggested as a causative factor for the development of insulin resistance and NAFLD through increasing oxidative stress via production of hydrogen peroxide (*Kelley et al., 2010*; *Xu et al., 2015*). Therefore, our independent metabolomics analyses confirm some of the main points of our study.

## Transcriptomics of aged mice suggest progression of the disease-like phenotype

To better understand changes happening in aged *Cdk1* cKO liver, we performed transcriptomic analysis on 12-month-old control and *Cdk1* cKO liver by RNA-seq, using equivalent parts of the liver. Principal component analysis of the transcriptomic data indicated that control and *Cdk1* cKO samples clustered separately along the PC1 axis, with the PC1 axis accounting for 68.7% of the variability among the samples, and that the transcriptomes of aged control samples were more similar to

each other while those of *Cdk1* cKO liver were more heterogeneous (*Figure 7A*). We identified a total of 12,143 genes, including 310 genes that were upregulated and 72 that were downregulated in *Cdk1* cKO liver relative to control (*Figure 7B*, *Supplementary file 6*). Some of the top upregulated were predicted genes with relatively unknown functions, such as *Gm14295*, *Gm11007*, and *Gm2007*, although we also found genes such as *Gsta1*, an oxidative stress response gene that is one of the most differentially expressed between normal liver and steatotic liver (*Hennig et al., 2014*), and retinol metabolism genes including *Cyp2b9*, *Cyp2b13*, and *Rdh9*, which is part of a predictive signature for fibrosis in a mouse model for NASH (*van Koppen et al., 2018*). Among the top

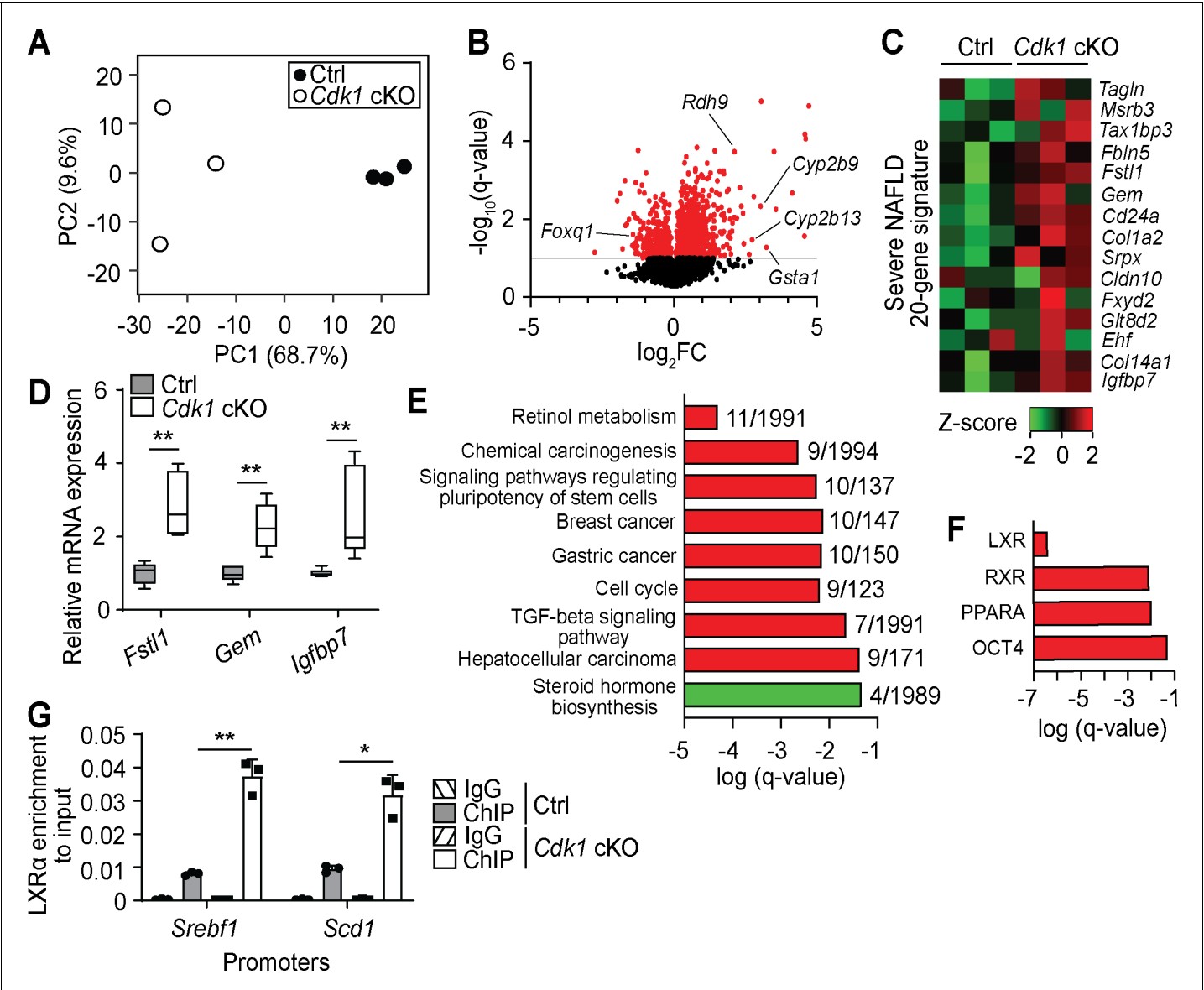

**Figure 7.** Transcriptomic analysis of aged *Cdk1* cKO mice. (**A**) Principal component analysis of RNA-seq data from 12-month-old mice. (**B**) Volcano plot of RNA-seq data. Significantly differentially expressed genes (q-value <0.1) are represented by red dots, while non-significant genes by black dots. (**C**) Heat map of expression data from RNA-seq of genes from a severe NAFLD 20-gene signature. RNA-seq data is available in *Supplementary file 6*. (**D**) qPCR validation of selected genes from the 20-gene signature (n = 5 per genotype). (**E**) KEGG pathway analysis of differentially expressed genes (red for upregulated genes, green for downregulated genes) from RNA-seq data. Number of genes over the total number of genes associated with each KEGG term is indicated at the end of the respective bars. (**F**) ChIP enrichment analysis (ChEA) of differentially expressed genes from RNA-seq data. KEGG and ChEA data are available in *Supplementary file 7*. (**G**) Enrichment of LXRα localization at promoters of *Srebf1* and *Scd1* upon ChIP-qPCR after normalization to input. Error bars for all graphs represent S.E.M. Source data for qPCR and ChIP-qPCR are available in *Supplementary file 10*.

downregulated genes, we identified *Foxq1*, an established repressor of FOXO1 (*Cui et al., 2016*), confirming the presence of insulin resistance in which FOXO1 is hyperactive. Interestingly, *Foxq1* was expressed at normal levels in the 8-week-old *Cdk1* cKO mice (see *Figure 1L*), indicating that with age, there is an adaptive process regulating FOXQ1/FOXO1 activity. When we compared the RNA-seq data to a 20-gene signature associated with severe NAFLD/NASH relative to mild NAFLD (*Moylan et al., 2014*), the gene signature appeared to be expressed substantially higher in aged *Cdk1* cKO liver than in aged control liver (*Figure 7C*). We further validated selected targets within the gene signature via qPCR and confirmed that these genes were upregulated in aged *Cdk1* cKO liver (*Figure 7D*).

KEGG pathway analysis (*Figure 7E*, *Supplementary file 7*) indicated a number of pathways are enriched among upregulated (red) and downregulated (green) genes from the RNA-seq. The most significantly enriched pathway, 'retinol metabolism', as well as enrichment of 'TGF-beta signaling pathway', provide support for the presence of fibrosis in *Cdk1* cKO liver (*Lee and Jeong, 2012*; *Meng et al., 2016*) and confirmed our metabolomics data (*Figure 6J*). Multiple KEGG terms associated with cancer, such as 'chemical carcinogenesis', 'cell cycle' and 'hepatocellular carcinoma' were also enriched for upregulated genes (*Figure 7E*). On the other hand, KEGG pathway analysis of downregulated genes only highlighted the term 'steroid hormone biosynthesis' (*Figure 7E*). Nevertheless, a closer inspection of genes classified under this term revealed *Hsd3b5* and *Cyp2c70* for which downregulation is associated with development of NAFLD in mouse models (*Hou et al., 2019*). ChIP enrichment analysis (ChEA) was also performed on the transcriptomic data to identify transcription factors whose downstream targets are enriched among differentially expressed genes (*Figure 7F*, *Supplementary file 7*). LXR, a glucose-responsive lipogenic nuclear receptor (*Mitro et al., 2007*), was the most significant transcription factor identified. LXR binds to DNA as a heterodimer with RXR (*Willy et al., 1995*), which also appears as an enriched transcription factor, indicating that the LXR-RXR heterodimer may be more active in the aged *Cdk1* cKO liver. Indeed, ChIP-qPCR for LXRα identified increased binding of LXRα at promoters of *Srebf1* and *Scd1* (*Figure 7G*; p=0.0093 for *Srebf1* promoter and p=0.0213 for *Scd1* promoter), both of which are lipogenic genes and are direct targets of LXR (*Repa, 2000*; *Chu et al., 2006*), suggesting that LXR might be directly responsible for the increase in SREBP1c levels (*Figure 6D–E*) and therefore lead indirectly to hepatic steatosis. Notably, OCT4, a transcription factor commonly associated with pluripotency in stem cells (*Chambers and Tomlinson, 2009*), was also selected by ChEA (*Figure 7F*). This concurs with the enrichment of the KEGG term 'signaling pathways regulating pluripotency of stem cells' (*Figure 7H*) and suggests potential OCT4-driven oncogenic de-differentiation of hepatocytes (*Yin et al., 2015*; *Sun et al., 2017*).

In conclusion, our findings suggest that upon loss of CDK1, hepatocytes become defective in FAO oxidation, causing excessive FFAs to promote hyperinsulinemia. Over time, chronic hyperinsulinemia culminates in the development of liver disease and a diabetes-like phenotype, as observed in aged *Cdk1* cKO liver by the presence of insulin resistance and a NASH-like phenotype that is potentially on course to oncogenesis (*Xu et al., 2017*).

## Discussion

In this study, we aimed to understand the impact of loss of CDK1 and the subsequent impairment of hepatocyte proliferation on lipid metabolism by performing lipidomics, metabolomics, and RNA-seq analyses on liver samples from our mouse model of defective hepatocyte proliferation, the *Cdk1* cKO mouse. One of the main observations was that TGs were reduced in the hepatocytes of *Cdk1* cKO mice when compared to age-matched control mice (*Figure 1A–D*). This is likely caused by increased oxidative stress-dependent induction of FOXO1-mediated transcription of *Pnpla2* (*Figure 1*), the gene coding for ATGL, the main enzyme involved in breaking down TGs to diacylglycerides (*Zimmermann et al., 2004*). Our findings contrast a previous study using the HepG2 hepatoma cell line, which showed that hydrogen peroxide-induced oxidative stress can instead lead to lipid accumulation via SREBP1c action (*Sekiya et al., 2008*). However, we believe this is because the study utilized very high levels of hydrogen peroxide to induce oxidative stress. Although physiological levels of oxidation can trigger SIRT1 activity leading to FOXO1 activation (*Alcendor et al., 2007*; *Prozorovski et al., 2008*; *Chakrabarti et al., 2011*), high levels of hydrogen peroxide can cause proteasome-mediated degradation of SIRT1 protein (*Yang et al., 2007*), thereby preventing the

SIRT1-FOXO1-ATGL pathway from being effective. Furthermore, since SIRT1 can inhibit the transcriptional capability of SREBP1c (*Ponugoti et al., 2010*), the loss of SIRT1 protein would also result in the derepression of SREBP1c.

Oxidative stress can induce FOXO1 activity through a number of different mechanisms (*Klotz et al., 2015*). For instance, oxidative stress may promote the dimerization of the RNA-binding protein HuR, which can, in turn, bind to the 3′ untranslated region and enhance stability of the FOXO1 mRNA (*Benoit et al., 2010*; *Li et al., 2013*). This would in part explain the increase in FOXO1 mRNA observed in young *Cdk1* cKO liver (*Figure 1L*). As discussed above, oxidative stress can also trigger SIRT1 activity by increasing the $NAD^+$/NADH ratio. SIRT1-dependent deacetylation of the FOXO1 protein can lead to nuclear trapping and increased FOXO1 localization on chromatin (*Frescas et al., 2005*), which might be demonstrated by the relatively elevated amounts of FOXO1 protein found at the *Pnpla2* promoter (*Figure 1O*) when compared to the increase in FOXO1 protein level in *Cdk1* cKO liver (*Figure 1M–N*).

Besides a reduction in TGs, we also observed an increase of ACs (*Figure 2A–B*). Through β-hydroxybutyrate tests and FAO assays (*Figure 2C–E*), we suggest that this is due to *Cdk1* cKO hepatocytes being deficient in FAO. This can happen as a result of a reduction in FAO enzymes (*Figure 2G–H*) or by decreasing SIRT3-mediated fatty acid import into the mitochondria (*Figure 2I–K*). Notably, both these mechanisms are dependent on the loss of CDK1 activity (*Harbauer et al., 2014*; *Liu et al., 2020*). With relevance to senescence, during which CDK activities are repressed to prevent cell cycle progression and maintain the irreversible cell cycle exit state, the loss of CDK1 function can lead to a block of FAO in hepatocytes. Indeed, it was previously shown that hepatocytes induced to senesce by irradiation end up with impaired mitochondrial β-oxidation (*Ogrodnik et al., 2017*). Hence, our findings provide a link between senescence and the loss of FAO in hepatocytes.

Impaired FAO, together with increased lipolysis, can lead to a buildup of FFAs. This can cause lipotoxicity and may explain a population of sub-$G_0$ hepatocytes seen in *Cdk1* cKO hepatocytes previously, in turn leading to immune infiltration and fibrosis (*Dewhurst et al., 2020*). Excess FFAs can also enter the bloodstream (*Figure 3A*) and be taken up and stored as TGs in the adipose tissue (*Figure 3H*) or modulate responses in other peripheral organs. In particular, increased levels of FFA in the blood enhances insulin secretion by pancreatic β-cells (*Figure 4B*). This is because β-cells express the fatty acid receptor GPR40 (*Itoh et al., 2003*), which, as a G-protein-coupled receptor, activates the phospholipase C pathway that triggers release of calcium ions from the endoplasmic reticulum and stimulates exocytosis of insulin from cytoplasmic insulin granules (*Rorsman and Ashcroft, 2018*; *Usui et al., 2019*). Alternatively, FFA can also promote upregulation of CYPD and increase the mitochondrial proton leak in β-cells, which enhances non-glucose-stimulated insulin secretion (*Taddeo et al., 2020*).

Being the main organ that carries out gluconeogenesis, the liver is one of the targets for insulin activity (*Petersen and Shulman, 2018*). Increased phosphorylation of INSRB, an insulin receptor subunit, and AKT (*Figure 4F–H*), a mediator of the insulin signaling pathway, reflects greater insulin signaling in young *Cdk1* cKO liver, in accordance with enhanced insulin secretion by the pancreas. Blood glucose levels were correspondingly decreased (*Figure 4D*) and hepatic glycogen storage was increased (*Figure 4E*). Paradoxically, FOXO1, which is normally phosphorylated and repressed by activated AKT upon insulin signaling (*Lu et al., 2012*), was hyperactive in *Cdk1* cKO liver (*Figure 1*). This discrepancy can be resolved by the notion that Sirtuin-dependent activation of FOXO1 takes precedence over mitogen-dependent repression of FOXO1, and that in the presence of oxidative stress, insulin-mediated FOXO1 phosphorylation is abrogated (*Frescas et al., 2005*).

While acute increases in insulin are important for maintaining blood glucose homeostasis, chronic insulin elevation, or chronic hyperinsulinemia, can lead to insulin resistance (*Morita et al., 2017*). This can happen by downregulation of the insulin receptor through miR-27b (*Srivastava et al., 2018*) or by constantly maintaining the insulin receptor in an 'insulin refractory state' whereby the insulin receptor has a lower ability to carry out auto-phosphorylation (*Catalano et al., 2014*). Notably, hyperinsulinemia was sustained upon aging in *Cdk1* cKO mice (*Figure 5A*) given that this is a constitutive knockout model. As a result, aged *Cdk1* cKO mice develop hepatic insulin resistance, likely through reduced INSRB expression (*Figure 5C–D*) and reduced insulin receptor auto-phosphorylation (*Figure 5E*).

The development of insulin resistance in aged *Cdk1* cKO mice is supported by our observation of reduced insulin signaling (*Figure 5C–F*), increased blood glucose levels (*Figure 5G*), reduced glucose tolerance (*Figure 5I–J*), and diminished response to insulin (*Figure 5K–L*). Analysis of our published metabolomics data from aged *Cdk1* cKO mice (*Narayanaswamy et al., 2020*) further reveals that purine metabolites were increased (*Figure 6L*). This is notable because an increase in purine metabolites, especially urate, is commonly observed in diabetic patients (*Papandreou et al., 2019*; *Varadaiah et al., 2019*). Nevertheless, the reason for this association between purine metabolites and insulin resistance remains unknown. One possibility is that upon insulin resistance, hyperglycemia, aggravated by hepatic gluconeogenesis and inhibition of glycogenesis, increases the flux through the pentose phosphate pathway (PPP) by increasing upstream metabolites. In the context of the *Cdk1* cKO mice, the presence of oxidative stress (*Figure 1*) may also exacerbate the situation by actively diverting glucose metabolites into the PPP to generate NADPH as the reducing cofactor for the oxidative stress response machinery (*Anastasiou et al., 2011*; *Kuehne et al., 2015*).

Insulin resistance is associated with hepatic steatosis, with about half of patients with type 2 diabetes also developing steatosis (*Roden, 2006*). Similarly, aged *Cdk1* cKO mice develop hepatic steatosis with an accumulation of TGs (*Figure 6A–B*), in contrast to young *Cdk1* cKO mice which have reduced amounts of TGs instead (*Figure 1*). This change can happen for a multitude of reasons. While we show diminished response to insulin in the liver (*Figure 5C–F*), one can expect that chronic hyperinsulinemia would result in insulin resistance in various peripheral organs such as the adipose tissue (*Figure 5—figure supplement 1*), leading to the inability of peripheral tissues to perform insulin-mediated glucose uptake. Furthermore, FOXO1 is already active in young *Cdk1* cKO mice, exacerbating hyperglycemia (*Figure 5G*) via gluconeogenesis. The increase in blood glucose levels can then trigger the activity of LXR (*Mitro et al., 2007*), which would in turn drive expression of *Srebf1* (*Figure 6D–E*, *Figure 7G*) and therefore lipogenesis. It is noteworthy that ChEA of RNA-seq data highlights LXR and RXR targets as being enriched among genes differentially expressed between *Cdk1* cKO and control mice (*Figure 7F*). As a result of increased LXR-dependent expression of lipogenic genes, the rate of lipogenesis likely increased over time in parallel with the progression of insulin resistance and eventually exceeded the rate of lipolysis. This would then lead to a net accumulation of lipids that caused the manifestation of steatosis, resulting in an inversion of the lipid phenotype in the liver of *Cdk1* cKO mice upon aging.

In addition to hepatic steatosis, we also observed increased fibrosis (*Figure 6G*) in the livers of aged *Cdk1* cKO mice. The presence of both these phenotypes is distinctive of NASH. Thus, it is unexpected that there was a lack of pro-inflammatory genes in the transcriptomics analysis of the aged mice (*Figure 7*). This would suggest that only the fibrotic aspect of NASH, but not the inflammatory aspect, was recapitulated. We hypothesize that this might be because hyperinsulinemia and insulin resistance mediate immune suppression (*Marín-Juez et al., 2014*; *Knuever et al., 2015*) by promoting the anti-inflammatory M2 phenotype in peripheral macrophages (*Ieronymaki et al., 2019*), such as the hepatic Kupffer cells, and by sensitizing macrophages to pro-apoptotic signals (*Senokuchi et al., 2008*). Despite the lack of an inflammatory response, liver fibrosis in aged *Cdk1* cKO mice could still be induced by factors released by steatotic hepatocytes that are sufficient to activate the fibrogenic hepatic stellate cells (*Wobser et al., 2009*).

CDK1 is primarily known as a driver of cell division, although it also has lesser known metabolic roles involving the regulation of mitochondrial function (*Harbauer et al., 2014*; *Liu et al., 2020*). As such, we cannot exclude the possibility that our observations are a result of loss of CDK1 activity instead of the general loss of hepatocyte proliferation. At this moment, it is unclear whether the cell cycle and metabolic functions of CDK1 are connected or can be uncoupled because there are no *Cdk1* mutants known that drive only one of these functions. We believe that it is likely that the impact on CDK1-associated metabolic processes contribute to the phenotypes, because our model of metabolic adaptation in *Cdk1* cKO mice (*Figure 8*) occurs primarily due to dysfunctional FAO in the hepatocytes, which tend to be quiescent (*Figure 2*). Hence, a better understanding of how loss of hepatocyte proliferation impacts liver physiology may entail further studies and comparisons with other animal models sporting impaired hepatocyte division. Nevertheless, there are many physiological and pathological instances where CDK1 activity is disrupted, and it is in these settings that findings from the *Cdk1* cKO mice might be clinically applicable. For example, during aging, CDK1 activity is reduced, partly due to reduced CDK1 expression and partly due to induction of CDK inhibitors upon senescence (*Stein et al., 1991*; *Wong and Riabowol, 1996*; *McConnell et al., 1998*;

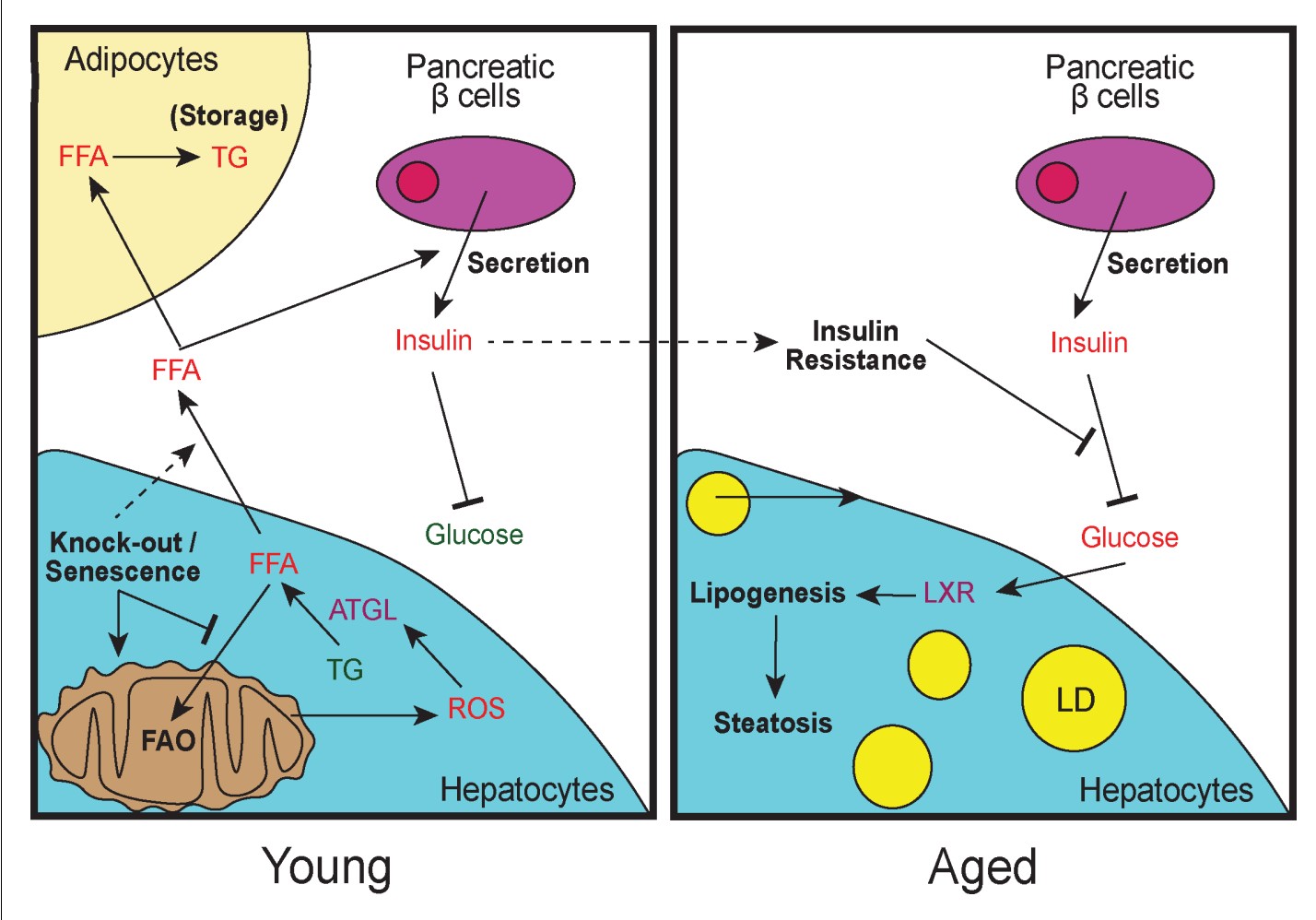

**Figure 8.** Model of lipid remodeling in young and aged *Cdk1* cKO mice. Upon knockout of *Cdk1* or senescence, whereby CDK1 activity is inhibited by CDK1 inhibitors, there is a block of FAO, leading to an accumulation of FFA in young *Cdk1* cKO mice. This is further exacerbated by mitochondrial oxidative stress that promotes FOXO1-dependent ATGL upregulation and increased lipolysis. The accumulated FFA then enter the bloodstream and is stored as TGs in adipocytes in the WAT. Elevated FFA levels in the bloodstream can also induce greater insulin secretion by pancreatic β-cells (hyperinsulinemia), leading to reduced blood glucose. However, over time, chronic hyperinsulinemia results in the development of insulin resistance in aged *Cdk1* cKO mice, which prevents the blood glucose lowering effect of insulin, causing hyperglycemia. Hyperglycemia, in turn, activates LXR activity and drives lipogenesis, eventually culminating in the manifestation of hepatic steatosis and a NAFLD-like phenotype.

*Quadri et al., 1998*), which may contribute to the reduction in hepatocyte proliferative capacity in aged liver (*Schmucker and Sanchez, 2011*). Correspondingly, our data suggest that attenuated CDK1 function might contribute to age-related hyperinsulinemia (*Kurauti et al., 2019*).

Meta-analyses of transcriptomics data from NAFLD patient livers have not identified CDK1 transcripts as being differentially expressed (*Ryaboshapkina and Hammar, 2017*; *Huang et al., 2018*), although aged *Cdk1* cKO mice develop steatotic livers. Still, hepatocytes from NAFLD patients display impaired proliferation, and with increasing evidence that senescence is rampant in the NAFLD liver (*Papatheodoridi et al., 2020*), it is not implausible to imagine that CDK1 is hypoactive in such conditions. Notably, *Ogrodnik et al., 2017* demonstrated that senescent hepatocytes contribute to development of hepatic steatosis as a result of defective mitochondrial FAO, and removal of these senescent hepatocytes ameliorates the phenotype. We provide a potential mechanistic pathway by which senescence can lead to steatosis, through inhibition of CDK1 activity, eventually leading to FFA-induced chronic hyperinsulinemia and the consequent insulin resistance. More recently, *Omori et al., 2020* uncovered that senescence is also increased in the hepatic non-parenchymal cell population. Furthermore, they confirmed that elimination of senescent cells rescues the NASH

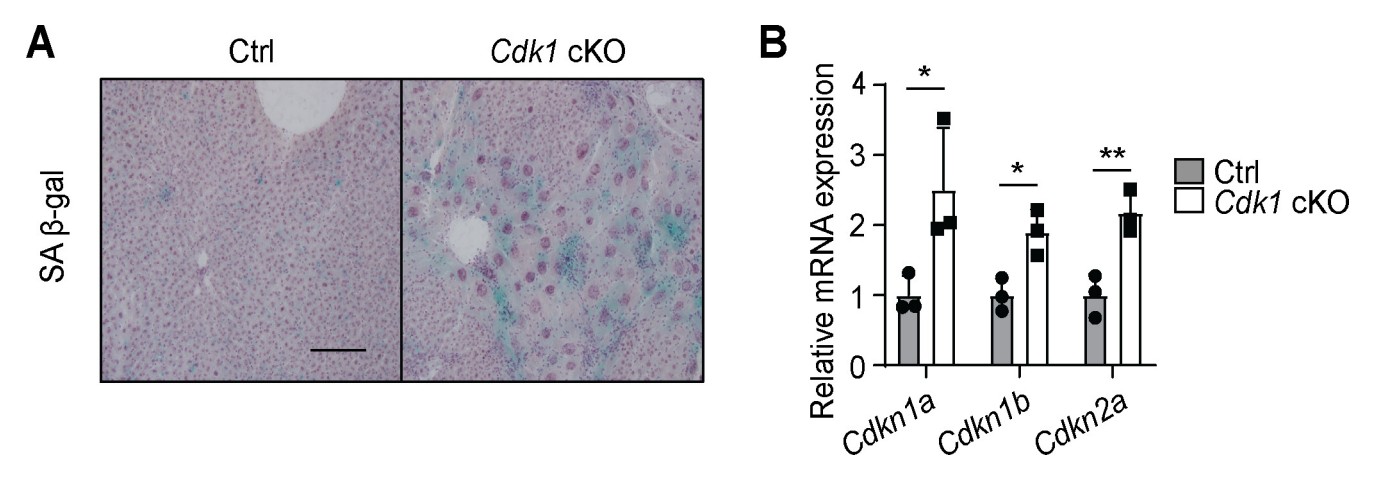

**Figure 9.** Senescence develops in young *Cdk1* cKO mice. (**A**) Representative image of senescence-associated β-galactosidase (SA β-gal) staining of liver sections from 8-week-old control (Ctrl) and *Cdk1* cKO mice. Scale bar represents 50 μm. (**B**) qPCR for *Cdkn1a* [p21$^{cip1/waf1}$], *Cdkn1b* [p27$^{Kip1}$], and *Cdkn2a* [p16$^{ink4a}$] in liver from 8-week-old Ctrl and *Cdk1* cKO mice (n = 3 per genotype). Data is available in ***Supplementary file 10***.

phenotype. Hence, it will be interesting to explore whether the deletion of CDK1 in the non-parenchymal cells also induces liver steatosis.

One foreseeable limitation of our model is that senescence is observed in the liver of young *Cdk1* cKO mice, evident from the presence of senescence-associated β-galactosidase staining (***Figure 9A***) and the induction of senescence markers such as *Cdkn1a* [p21$^{cip1/waf1}$], *Cdkn1b* [p27$^{Kip1}$], and *Cdkn2a* [p16$^{ink4a}$] (***Figure 9B***), in agreement with our previous findings (***Dewhurst et al., 2020***). This suggests the possibility that non-CDK1-dependent senescent pathways, such as the senescence-associated secretory phenotype (SASP), may contribute to hepatic steatosis upon aging. A rescue of mitochondrial FAO in *Cdk1* cKO mice would help us delineate the relative contribution of the FFA-hyperinsulinemia pathway to the phenotypes seen in the aged mice. Another limitation of the *Cdk1* cKO mice is that senescence typically occurs in a mosaic fashion (***Herbig et al., 2003***), while knock-out of CDK1 occurs in all hepatocytes in the *Cdk1* cKO mouse (***Dewhurst et al., 2020***). As such, not all the phenotypes seen in the aged *Cdk1* cKO mice might actually manifest during normal aging.

In this study, using our mouse model in which hepatocytes lack CDK1, the *Cdk1* cKO mouse, we show that the liver, in the absence of any external stimuli, develops metabolic disease upon aging, which also affects other tissues. With these findings, our study proposes that the loss of CDK1 activity in hepatocytes, in addition to being an outcome of liver disease, can be a contributor to hepatic pathology and supports the concept that senotherapeutic drugs targeting senescent cells (***Ritschka et al., 2020***) are potentially viable therapeutic options for treating liver metabolic diseases.

# Materials and methods

### Key resources table

| Reagent type (species) or resource | Designation | Source or reference | Identifiers | Additional information |
|---|---|---|---|---|
| Gene (*Mus musculus*) | *Cdk1* | NCBI Gene | ID:12534 | |
| Gene (*Mus musculus*) | *Cpt2* | NCBI Gene | ID:12896 | |
| Gene (*Mus musculus*) | *Foxo1* | NCBI Gene | ID:56458 | |

*Continued on next page*

*Continued*

| Reagent type (species) or resource | Designation | Source or reference | Identifiers | Additional information |
|---|---|---|---|---|
| Gene (*Mus musculus*) | *Pnpla2* | NCBI Gene | ID:66853 | |
| Strain, strain background (*Escherichia coli*) | Stbl3 | Thermo Fisher Scientific | Cat#:C7373-03 | For cloning and maintaining lentiviral vectors |
| Genetic reagent (*Mus musculus*) | Alb-Cre | DOI: 10.1002/(sici)1526-968x(200002)26:2 < 149::aid-gene 16 > 3.0.co;2 v | RRID:MGI:6258240 | Control to the *Cdk1* cKO mice |
| Genetic reagent (*Mus musculus*) | *Cdk1* cKO | DOI: 10.1073/pnas. 1115201109 | RRID:MGI:5318109 | Hepatocyte-specific knockout of *Cdk1* gene |
| Transfected construct (*Mus musculus*) | pLKO.1 | Addgene | Cat#:#8453; RRID:Addgene_10878 | Empty vector |
| Transfected construct (*Mus musculus*) | pLKO-shPnpla2 | This paper | | Expression of shRNA against *Pnpla2* (sequence: GGAGA GAACGTCATCATAT) |
| Biological sample (*Mus musculus*, male) | Liver, subcutaneous WAT and epididymal WAT from *Cdk1* cKO mice | This paper | | Freshly isolated from 2-, -6, or 12-month-old *Mus musculus* |
| Biological sample (*Mus musculus*, male) | Plasma from *Cdk1* cKO mice | This paper | | Freshly isolated from 2, 6, or 12 month old *Mus musculus* |
| Antibody | Anti-ACADVL (Mouse monoclonal) | Santa Cruz Biotechnology | Cat#:sc-376239; RRID:AB_10989696 | WB (1:1000) |
| Antibody | Anti-AKT (Rabbit polyclonal) | Cell Signaling | Cat#:#9272; RRID:AB_329827 | WB (1:2000) |
| Antibody | Anti-pT308-AKT (Rabbit polyclonal) | Cell Signaling | Cat#:#9275; RRID:AB_329829 | WB (1:500) |
| Antibody | Anti-ATGL (Mouse monoclonal) | Santa Cruz Biotechnology | Cat#:sc-365278; RRID:AB_10859044 | WB (1:2000) |
| Antibody | Anti-CPT2 (Rabbit polyclonal) | Proteintech | Cat#:26555–1-AP; RRID:AB_2880551 | IP (2 µL per 500 µg protein), WB (1:2000) |
| Antibody | Anti-FOXO1 (Rabbit monoclonal) | Cell Signaling | Cat#:#2880; RRID:AB_2106495 | ChIP (2 µL per 10 µg chromatin), WB (1:1000) |
| Antibody | Anti-HADHA (Mouse monoclonal) | Santa Cruz Biotechnology | Cat#:sc-374497; RRID:AB_10987868 | WB (1:500) |
| Antibody | Anti-HSP90 (Mouse monoclonal) | BD Transduction Laboratories | Cat#:610418; RRID:AB_397798 | WB (1:10,000) |
| Antibody | Anti-INSRB (Rabbit polyclonal) | Santa Cruz Biotechnology | Cat#:sc-711; RRID:AB_631835 | WB (1:2000) |
| Antibody | Anti-pY1151/ 1152-INSRB (Mouse monoclonal) | Santa Cruz Biotechnology | Cat#:sc-81500; RRID:AB_1125642 | WB (1:500) |
| Antibody | Anti-LXRα (Mouse monoclonal) | Abcam | Cat#:ab41902; RRID:AB_776094 | ChIP (2 µL per 10 µg chromatin) |
| Antibody | Anti-pan-acetylation (Mouse monoclonal) | Proteintech | Cat#:66289–1-Ig; RRID:AB_2881672 | WB (1:1000) |

*Continued on next page*

*Continued*

| Reagent type (species) or resource | Designation | Source or reference | Identifiers | Additional information |
|---|---|---|---|---|
| Antibody | Anti-phospho-Ser/Thr (Rabbit polyclonal) | Abcam | Cat#:ab117253; RRID:AB_10903259 | WB (1:4000) |
| Antibody | Anti-SIRT3 (Rabbit monoclonal) | Cell Signaling | Cat#:#5490; RRID:AB_10828246 | IP (2 µL per 500 µg protein), WB (1:2000) |
| Antibody | Anti-SREBP1c (Rabbit polyclonal) | Abcam | Cat#:ab28481; RRID:AB_778069 | WB (1:1000) |
| Peptide, recombinant protein | Actrapid, fast-acting insulin | Novo Nordisk | | |
| Peptide, recombinant protein | rDNase | Macherey-Nagel | Cat#:740963 | |
| Commercial assay or kit | Triglyceride Assay Kit | Abcam | Cat#:ab65336 | |
| Commercial assay or kit | HDL and LDL/VLDL Cholesterol Assay Kit | Abcam | Cat#:ab65390 | |
| Commercial assay or kit | Free Fatty Acid Assay Kit | Cell Biolabs | Cat#:STA-618 | |
| Commercial assay or kit | β-Hydroxybutyrate (Ketone Body) Colorimetric Assay Kit | Cayman Chemicals | Cat#:700190 | |
| Commercial assay or kit | Ultra Sensitive Mouse Insulin ELISA Kit | Crystal Chem | Cat#:90080 | |
| Commercial assay or kit | Glycogen Assay Kit | Abcam | Cat#:ab83369 | |
| Commercial assay or kit | Fatty Acid Oxidation Assay Kit | Abcam | Cat#:ab217602 | |
| Commercial assay or kit | Extracellular Oxygen Consumption Assay Kit | Abcam | Cat#:ab197243 | |
| Commercial assay or kit | Maxima First Strand cDNA Synthesis Kit | Thermo Fisher Scientific | Cat#:K1641 | |
| Commercial assay or kit | Maxima SYBR Green qPCR Master Mix | Thermo Fisher Scientific | Cat#:K0221 | |
| Commercial assay or kit | PureLink RNA Mini Kit | Thermo Fisher Scientific | Cat#:12183025 | |
| Commercial assay or kit | Oxidized Protein Western Blot Detection Kit | Abcam | Cat#: ab178020 | |
| Chemical compound, drug | N-acetylcysteine | Sigma-Aldrich | Cat#:A7250 | |
| Software, algorithm | GraphPad Prism | GraphPad | RRID:SCR_002798 | Version 6 |
| Software, algorithm | STAR-mapper | DOI: 10.1093/bioinformatics/bts635 | | For RNA-seq analysis |

*Continued on next page*

Continued

| Reagent type (species) or resource | Designation | Source or reference | Identifiers | Additional information |
|---|---|---|---|---|
| Software, algorithm | RSEM | DOI:10.1186/1471-2105-12-323 | RRID:SCR_013027 | For RNA-seq analysis |
| Software, algorithm | ClustVis | DOI:10.1093/nar/gkv468 | RRID:SCR_017133 | For RNA-seq analysis |
| Software, algorithm | Enrichr | DOI:10.1093/nar/gkw377 | RRID:SCR_001575 | For RNA-seq analysis |
| Software, algorithm | MetaboKit | DOI:10.1039/d0mo00030b | | For untargeted metabolomics analysis |
| Other | Accu-Chek Performa Nano | Accu-Chek | | Blood glucose meter |
| Other | Accu-Chek test strips | Accu-Chek | Code 222 | Test strips for blood glucose meter |

## Genetic mouse models, animal experiments, and blood glucose test

*Cdk1* cKO mice have been previously described (*Diril et al., 2012*; *Caldez et al., 2018*). Briefly, *Cdk1* cKO were established by crossing *Cdk1^{fl/fl}* mice with Albumin-Cre mice (*Postic and Magnuson, 2000*; #003574, The Jackson Laboratory) for deletion of the *Cdk1* gene specifically in hepatocytes. The *Cdk1^{fl/fl}* Albumin-Cre mice will be designated as '*Cdk1* cKO' throughout this study. Hence, *Cdk1^{+/+}* Alb-Cre mice were used as controls (Ctrl) to account for expression of the Cre recombinase in the hepatocytes. Mice were maintained on standard chow ad libitum under 12 hr light/dark cycle. Only male mice were used for experiments to avoid hormonal confounding. For food intake measurement, 5-week-old mice were individually housed and given 1 week to acclimatize, after which the amount of food eaten was measured for 2 weeks (6- to 8-week-old) as described previously (*Bachmanov et al., 2002*). For N-acetylcysteine (NAC; A7250, Sigma-Aldrich) treatment, 6-week-old mice were fed with water containing 2 mg/mL NAC ad libitum for 2 weeks. As controls, mice were fed with normal water as usual. For *Pnpla2* knockdown in the liver, 40 µg of pLKO-shPnpla2 or empty vector were injected into 7-week-old mice by hydrodynamic tail vein injection (*Yokoo et al., 2016*). In brief, plasmids were diluted in Ringer's lactate solution at a volume corresponding to 10% body weight of mice and injected into the lateral tail vein within 10 s. Tissues were collected at the indicated ages. Blood was collected by cardiac puncture, transferred to a lithium heparin-coated Microvette 500 LH (20.1345.100, Sarstedt) and subsequently centrifuged at 14,000 rpm for 10 min at 4°C to collect plasma. Whole body fat mass was measured on live mice using EchoMRI Body Composition Analyzer (EchoMRI). Glucose tolerance tests (GTT) were performed by fasting mice for 16 hr followed by intraperitoneal injection of D-glucose at 1 g/kg body weight. Blood glucose was measured before injection and at 15, 30, 60, 90, and 120 min post-injection. For insulin tolerance tests (ITT), mice were fasted for 6 hr followed by intraperitoneal injection of insulin (Actrapid, Novo Nordisk) at 1 U/kg body weight and blood glucose was measured as above. Blood glucose measurements were performed on blood from tail snips using the Accu-Chek Performa Nano glucose meter (Roche) and Accu-Chek Performa test strips Code 222. Area under curve (AUC; for GTT) or area above curve (for ITT) was taken as the area above or below baseline (blood glucose level at 0 min), respectively, and calculated using GraphPad Prism version 6. All animal experiments were performed in accordance to protocols (#171268) approved by the A*STAR Institutional Animal Care and Use Committee (IACUC) based on the National Advisory Committee for Laboratory Animal Research (NACLAR) Guidelines.

## Biochemical assays

Hepatic and plasma TGs were measured using the Triglyceride Assay Kit (ab65336, Abcam), plasma cholesterols using the HDL and LDL/VLDL Cholesterol Assay Kit (ab65390, Abcam), plasma FFAs using FFA Assay Kit (STA-618, Cell Biolabs), hepatic β-hydroxybutyrate using β-hydroxybutyrate (Ketone Body) Colorimetric Assay Kit (700190, Cayman Chemicals), plasma insulin using the Ultra-

Sensitive Mouse Insulin ELISA Kit (90080, Crystal Chem), and hepatic glycogen using the Glycogen Assay Kit (ab83369, Abcam) according to the manufacturer's protocols. Signals were read using a TECAN Safire microplate reader at default parameters.

## Molecular cloning

shRNA targeting murine *Pnpla2* with the sequence 5'-GGAGAGAACGTCATCATAT-3' (*Miyoshi et al., 2007*) were designed and inserted into the pLKO.1 vector as previously published (*Moffat et al., 2006*) to generate the pLKO-shPnpla2 plasmid for knocking down *Pnpla2*. Plasmids were maintained and amplified in Stbl3 *Escherichia coli* cells (Thermo Fisher Scientific) and purified using EndoFree Plasmid Maxi Kit (Qiagen) before being used for hydrodynamic tail vein injection.

## β-oxidation assays

Hepatocytes were isolated and cultured as described previously (*Caldez et al., 2018*). β-oxidation capabilities of isolated hepatocytes were then measured using the FFA Assay Kit (ab217602, Abcam) in combination with the Extracellular Oxygen Consumption Assay Kit (ab197243, Abcam) following manufacturer's protocol. Signals were read using a TECAN Safire microplate reader with parameters as indicated in the assay protocol.

## Lipid extraction from liver tissue and plasma samples

Frozen liver tissues were lyophilized until constant dry weight in a vacuum concentrator. PBS (phosphate buffered saline) was added to the dried samples (30 µL per mg dry weight) and homogenized with homogenization beads using an Omni beadruptor homogenizer (speed: 3.50 m/s; cycle: six times; duration: 45 s, dwell time: 15 s, repeat four times). Homogenates were transferred to clean polypropylene tubes for lipid extraction. For lipid extraction, 20 µL of tissue homogenates were resuspended in 360 µL of chilled chloroform/methanol (1:2 v/v) containing internal standards. The samples were then placed on a rotary shaker (30 min, 4°C, 700 rpm). 120 µL of chilled chloroform and 100 µL of chilled milliQ water were then added, the samples vortexed again for 15 s, and centrifuged at 10,000 rpm for 7 min to separate the phases. The lower organic phases were collected and the remaining aqueous phases were re-extracted with 500 µL of chilled chloroform as above. The lower organic phases were combined with the first organic extracts. Lipid extracts were then dried in a vacuum concentrator, resuspended in chloroform/methanol (1:1 v/v) and kept at −80°C until LC-MS/MS analysis. Lipid extraction from 10 µL of plasma was performed as described above using half the volume of indicated reagents.

## Liquid chromatography with tandem mass spectrometry (LC-MS/MS) analysis of TGs, PLs, and ACs

Chromatography separation of TGs was achieved by RPLC on an Agilent Eclipse Plus C18 (100 × 2.1 mm, 1.8 µm), using an Agilent 1290 Infinity II LC system. The column temperature was 40°C, the autosampler was kept at 8°C, and 1 µL of sample was injected. Solvent A was acetonitrile/water (4:6 v/v), solvent B was acetonitrile/isopropanol (1:9 v/v), both solvent A and solvent B contained 10 mM ammonium formate. Gradient elution started at 20% solvent B, increased linearly to 60% in 2 min, then increased linearly to 100% solvent B in 10 min, held at 100% solvent B for 2 min, then brought back to 20% solvent B and held for 1.8 min (total run-time 15.8 min). The flow rate was 400 µL/min. The column effluent was introduced into an Agilent 6490 Triple Quadrupole MS system equipped with an electrospray ion source. MS parameters were as follows: Gas Temperature, 200°C; Gas Flow, 15 L/min; Nebulizer, 25 psi; Sheath Gas Heater, 250°C, Sheath Gas Flow, 12 L/min; Capillary, 3.5kV. TGs were measured in positive ionization, using both dynamic multiple reaction monitoring (dMRM) and single ion monitoring (SIM) (see *Supplementary file 8* for MRM lists). Data analysis was performed using Agilent MassHunter Quantitative Analysis (QQQ) software. The data were inspected manually to ensure peak integration by the software was appropriate. AUC of the integrated ion chromatogram peaks for each MRM transition were extracted to Microsoft Excel and normalized to the AUC of the d5-TG 48:0 internal standard.

Chromatography separation of PL and AC was achieved by hydrophilic interaction liquid chromatography (HILIC) on a Phenomenex Kinetex HILIC column (150 × 2.10 mm, 2.6 µM, 100 Å), using an Agilent 1290 Infinity II LC system. The column temperature was 50°C, the autosampler was kept at 8°

C, and 1 µL of sample was injected. Solvent A and Solvent B were both acetonitrile/25 mM aqueous ammonium formate pH4.6 with distinct ratios (1:1 v/v and 19:1 v/v, respectively). Gradient elution started at 1% solvent A, increased linearly to 25% solvent A in 6 min, then increased linearly to 90% solvent A in 1 min, then decreased back at 1% solvent A in 0.1 min, and held for 3 min (total runtime 10.1 min). The flow rate was 500 µL/min. For phosphatidylserine analysis, a slightly different gradient was used: elution started at 1% solvent A, increased linearly to 75% solvent A in 6 min, then increased linearly to 90% A in 1 min, then decreased back at 1% A in 0.1 min, and held for 3 min; total runtime (10.1 min). The column effluent was subjected to MS and data analysis as described for TG with minor changes as described below. PLs and ACs were measured in positive or negative ionization, using multiple reaction monitoring (MRM) (see *Supplementary file 8* for MRM lists). Isotopic correction of AUC of the PLs and sphingomyelin was done using in-house R script. Corrected AUC were normalized to the AUC of class-specific internal standards.

For quality control, prior to lipid extraction, samples were randomized (using Excel random number generator) and quality control (QC) samples were prepared by pooling aliquots from each sample indicated by randomization. QC samples were used to monitor reproducibility of extraction, linearity and stability of instrument response. Blank extracts were also prepared by adding the extraction mixture to empty tubes, and used to monitor carry-over and contamination issues. Briefly, MRM transitions were kept for analysis only if they satisfied the following criteria: RSD <25% over all QC samples, linearity curve $R^2$ >0.9, and signal in the blank less than 10% of average signal in the QC sample. Heat maps were generated using Heatmapper (*Babicki et al., 2016*).

## Non-esterified FFA extraction and LC-MRM analysis

FFAs were analyzed as described (*Christinat et al., 2016*). Briefly, 50 µL of plasma was mixed with 450 µL of isopropanol containing 5 nmol internal standard palmitic acid-d4 (Sigma Aldrich) in 2 mL polypropylene tubes, vortexed for 5 s, and placed in a rotary shaker (30 min, 4°C, 700 rpm). The samples were then centrifuged (10 min, 15,000 rpm, 4°C) and 200 µL of supernatant was transferred to a new 2 mL tube. The solvent was evaporated in a vacuum centrifuge, and the samples were reconstituted in 100 µL of acetonitrile/water (1:1 v/v), transferred to an autosampler vial, and stored at −80°C. On the day of analysis, samples were thawed at room temperature for 30 min, vortexed for 5 s, sonicated for 1 min, and centrifuged for 5 min prior to LC/MS analysis.

Chromatographic separation of FFA was achieved by reverse phase liquid chromatography (RPLC) on a Waters Acquity CSH C18 150 × 2.1 mm column, in a Thermo Vanquish UHPLC system. The column temperature was 55°C, the autosampler was kept at 10°C, and 1 µL of sample was injected. Solvent A was acetonitrile/water (3:2 v/v), solvent B was acetonitrile/isopropanol (1:1 v/v), both solvent A and solvent B contained 10 mM ammonium acetate. Gradient elution started at 10% solvent B, which was held for 2 min, then increased linearly to 46% solvent B over 12 min, then to 100% solvent B over 3.5 min. The column was then flushed for 3 min with 100% solvent B and re-equilibrated under starting conditions for 3.5 min. The flow rate was 450 µL/min. The column effluent was introduced into a QExactive Plus quadrupole-orbitrap mass spectrometer via a HESI II ion source operating under the following conditions: spray voltage, 3 kV; capillary temperature, 350°C; sheath gas, 35 arbitrary units; aux gas, 10 arbitrary units; probe heater, 300°C; S-lens RF level, 50. Automatic gain control was set to 3E6 ions to enter the mass analyzer with a maximum ion time of 200 ms. Negative full scan profile spectra were acquired from 110 to 380 m/z at a resolution setting of 140,000 (FWHM at *m/z* 200). Data analysis was performed with Xcalibur Qual Browser, using the areas of extracted ion count chromatograms of deprotonated NEFAs with a mass tolerance of 5 ppm. Quantification was based on a one-point calibration with the internal standard palmitic acid d4.

## RNA isolation and quantitative real-time PCR (qPCR)

RNA was isolated from tissues using TRIzol reagent (Thermo Fisher Scientific) as per provided instructions, with an initial step of homogenizing the tissues in TRIzol reagent within a bead-containing Lysing Matrix D tube (MP Biomedicals) using the Precellys 24 homogeniser (Bertin Technologies). Complementary DNA (cDNA) was prepared from 2 µg of RNA with the Maxima First Strand cDNA Synthesis Kit (K1641, Thermo Fisher Scientific). qPCR was ran using the Maxima SYBR Green qPCR Master Mix (K0221, Thermo Fisher Scientific) with 10 ng of cDNA per reaction. Analysis was done

with the $2^{-\Delta\Delta Ct}$ method (*Livak and Schmittgen, 2001*) using *Eef2* as loading control. Primers used are provided in *Supplementary file 9*.

## RNA sequencing (RNA-seq), data processing, and analysis

For RNA extraction, tissue was homogenized and lysed with TRIzol as described above. Upon phase separation, the aqueous phase was transferred to spin columns from the PureLink RNA Mini Kit (12183025, Thermo Fisher Scientific) and washed as instructed by the kit manual. DNase treatment (740963, Macherey-Nagel) was done on-column before eluting with nuclease-free water. Subsequent RNA fragmentation, library generation and sequencing were as described previously (*Caldez et al., 2018*). For data analysis, sequence read alignment was done with STAR-mapper (*Dobin et al., 2013*) and fragments per kilobase million (FPKM) quantification with RSEM (*Li and Dewey, 2011*). FastQC was used to perform pre-alignment sequence read quality control, and upon inspection of the FPKM data, we decided not to do data normalization given the equal distribution of FPKM values across samples. A small fudge factor of 0.5 was added to all FPKM values to avoid artefacts associated with statistical analysis of extremely low abundance genes. Principal component analysis was performed using ClustVis (*Metsalu and Vilo, 2015*). Differential expression analysis was performed by two sample T-tests and multiple testing corrected by calculating q-value (*Storey, 2002*). Genes with q-value of <0.05 and fold change of >1.5 or <0.66 were considered significantly differentially expressed. KEGG pathway and ChEA of these genes were then done using Enrichr (*Kuleshov et al., 2016*). Raw sequencing data is available at NCBI GEO under accession number GSE159498 (https://www.ncbi.nlm.nih.gov/geo/query/acc.cgi?acc=GSE159498).

## Untargeted metabolomics of whole liver

The semi-quantitative metabolomics data for 870 unique compounds has been published in *Narayanaswamy et al., 2020*. We reprocessed the raw data with the latest version of MetaboKit software for improved accuracy in peak integration (August 2020). Metabolites with reliable quantification in at least three samples were used for analysis.

## Immunoprecipitation and immunoblotting

Proteins were extracted from tissues in RIPA buffer (50 mM Tris-HCl pH8.0, 50 mM NaCl, 1 mM EDTA, 1% NP-40, 0.1% sodium deoxycholate, supplemented with protease inhibitor) by homogenising in a pestle homogenizer, sonicated at high power for three cycles of 30 s ON/OFF on Bioruptor sonicator (Diagenode), and centrifuged at maximum speed for 10 min to remove cell debris. Proteins were then quantified using BCA Protein Assay (Thermo Fisher Scientific). Lysates were separated on polyacrylamide gels and transferred to PVDF membranes using the Bio-Rad Mini-Protean system. Probing and development of membrane were done as previously described (*Caldez et al., 2018*). Antibodies used for immunoblotting are provided in the Key Resources Table. Quantification of blots was performed with Fiji software (*Schindelin et al., 2012*). For detecting carbonylated proteins, 2,4-dinitrophenylhydrazine (DNP) derivation and the subsequent immunoblotting was performed using the Oxidized Protein Western Blot Detection Kit (ab178020, Abcam) following manufacturer's instructions.

For immunoprecipitation, 500 µg of pre-cleared protein lysate was incubated with 2 µL of anti-SIRT3 (#5490, Cell Signaling) or anti-CPT2 (26555–1-AP, Proteintech) antibody overnight in IP buffer (50 mM Tris-HCl pH8.0, 150 mM NaCl, 0.15% NP-40, 10% glycerol) in the cold. Antibodies were captured with Protein A agarose beads (15918–014, Thermo Fisher Scientific) on rotation for 1 hr the following day, and washed with IP buffer thrice for 5 min each. Beads were then incubated with SDS loading dye for 5 min at 95°C with shaking before being loaded on a polyacrylamide gel for immunoblotting as described above.

## Chromatin immunoprecipitation

Isolated hepatocytes were fixed in 1% formalin in suspension for 10 min at room temperature and then quenched in 125 mM glycine. Cells were pelleted and washed twice with cold PBS, and subsequent steps of sonication and immunoprecipitation were performed as described (*Shang et al., 2002*). Two µL of anti-FOXO1 (#2880, Cell Signaling) or anti-LXRα antibody (ab41902, Abcam) was used per ChIP sample. Primers used for ChIP-qPCR are listed in *Supplementary file 9*.

## Histology

H&E staining and Oil Red O staining were performed as described (*Niska-Blakie et al., 2020*), and senescence-associated β-galactosidase staining as previously published (*Diril et al., 2012*). For Sirius Red staining, slides containing paraffin-embedded tissue were deparaffinized and rehydrated at room temperature. Slides were then incubated in Picro-Sirius Red solution (ab246832, Abcam) for 4 hr, washed in running water and dehydrated before being mounted with Eukitt Quick-hardening mounting medium (03989, Sigma-Aldrich). Images were taken using the Olympus BX-61 upright microscope with 20X air or 40X oil lens. Histopathological analysis of liver H and E and Sirius Red-stained sections was performed and scored by a qualified pathologist (C.B.O.) using a scale of 0–5 [0 – no abnormalities detected; 1 - minimal (<1%); 2 – mild (1–25%); 3 – moderate (26–50%); 4 – marked (51–75%); 5 – severe (76–100%)] as described previously (*Shackelford et al., 2002*). Quantification of Sirius-Red-positive areas was done using Fiji (*Schindelin et al., 2012*). Measurement of adipocyte size was done with Adobe Photoshop CC 2018 using the Quick Selection tool to outline individual adipocytes followed by the Record Measurement function after the scale had been set.

## Statistical analysis

Statistical analyses were performed using Mann-Whitney non-parametric test, or unpaired two-tailed T-test with Welch's correction for experiments involving isolated hepatocytes, immunoblot quantification and ChIP-qPCR, on GraphPad Prism version 6, and were considered significant when p-value<0.05. Statistical significance was indicated as: p-value<0.05 (*); p-value<0.01 (**); p-value<0.001 (***). Correlation analysis to derive Pearson's correlation coefficient (r) and goodness-of-fit coefficient ($R^2$) values was also performed using GraphPad Prism version 6.

# Acknowledgements

We thank all present and past members of the Kaldis laboratory for discussions, input, and support. Special thanks go to Tan Qing Hui, Shiela Fransisco Margallo, Pangalingan Christie Chrisma Domingo, and Timothy Teck Chiew Chua for animal expertise. We acknowledge the Advanced Molecular Pathology Laboratory (AMPL) at the Institute of Cell and Molecular Biology (IMCB) for providing technical expertise support during histology processing.

# Additional information

## Funding

| Funder | Grant reference number | Author |
|---|---|---|
| Agency for Science, Technology and Research | IAF-ICP I1901E0040 | Amaury Cazenave-Gassiot Markus R Wenk |
| Singapore International Graduate Award | | Gözde Zafer |
| National Research Foundation Singapore | NRF2016-CRP001-103 | Philipp Kaldis |
| National Medical Research Council | NMRC-CG-M009 | Hyungwon Choi |
| National University of Singapore | Life Sciences Institute | Juat Chin Foo Amaury Cazenave-Gassiot Markus R Wenk |
| Swedish Foundation for Strategic Environmental Research | Dnr IRC15-0067 | Philipp Kaldis |
| Swedish Research Council | Dnr 2009-1039 | Philipp Kaldis |

The funders had no role in study design, data collection and interpretation, or the decision to submit the work for publication.

## Author contributions
Jin Rong Ow, Conceptualization, Data curation, Formal analysis, Validation, Investigation, Visualization, Methodology, Writing - original draft, Writing - review and editing; Matias J Caldez, Conceptualization, Data curation, Formal analysis, Validation, Investigation, Methodology, Writing - review and editing; Gözde Zafer, Data curation, Investigation, Methodology; Juat Chin Foo, Data curation, Methodology; Hong Yu Li, Heike Wollmann, Data curation, Formal analysis, Investigation, Methodology; Soumita Ghosh, Software, Formal analysis, Validation, Methodology; Amaury Cazenave-Gassiot, Resources, Data curation, Formal analysis, Supervision, Methodology; Chee Bing Ong, Formal analysis, Visualization, Methodology; Markus R Wenk, Weiping Han, Resources, Supervision; Hyungwon Choi, Resources, Software, Formal analysis, Supervision, Validation, Visualization, Methodology, Writing - review and editing; Philipp Kaldis, Conceptualization, Resources, Formal analysis, Supervision, Funding acquisition, Investigation, Writing - original draft, Project administration, Writing - review and editing

## Author ORCIDs
Jin Rong Ow ![ORCID] http://orcid.org/0000-0002-7468-691X
Weiping Han ![ORCID] http://orcid.org/0000-0002-5023-2104
Hyungwon Choi ![ORCID] http://orcid.org/0000-0002-6687-3088
Philipp Kaldis ![ORCID] https://orcid.org/0000-0002-7247-7591

## Ethics
Animal experimentation: All animal experiments were performed in accordance to protocols (#171268) approved by the A*STAR Institutional Animal Care and Use Committee (IACUC) based on the National Advisory Committee for Laboratory Animal Research (NACLAR) Guidelines.

## Decision letter and Author response
Decision letter https://doi.org/10.7554/eLife.63835.sa1
Author response https://doi.org/10.7554/eLife.63835.sa2

# Additional files

## Supplementary files
• Supplementary file 1. Lipidomics analysis of liver of 8-week-old control and *Cdk1* cKO mice.

• Supplementary file 2. Lipidomics analysis of hepatocytes isolated from 8-week-old control and *Cdk1* cKO mice.

• Supplementary file 3. Free fatty acid analysis of plasma from 8-week-old control and *Cdk1* cKO mice.

• Supplementary file 4. Histopathological scoring of H and E liver sections from 12-month-old control and *Cdk1* cKO mice.

• Supplementary file 5. Lipidomics analysis of liver of 12-month-old control and *Cdk1* cKO mice.

• Supplementary file 6. List of total detected and differentially expressed genes from RNA-seq data from liver of 12-month-old control and *Cdk1* cKO mice.

• Supplementary file 7. List of KEGG terms for up- and downregulated differentially expressed genes and list of transcription factors from ChIP enrichment analysis.

• Supplementary file 8. MRM list for mass spectrometry lipidomics analysis.

• Supplementary file 9. List of primers used for qPCR.

• Supplementary file 10. Source data used for all figures.

• Transparent reporting form

## Data availability
Raw sequencing data is available at NCBI GEO under accession number GSE159498.

The following dataset was generated:

| Author(s) | Year | Dataset title | Dataset URL | Database and Identifier |
|---|---|---|---|---|
| Ow J, Caldez MJ, Ghosh S, Wollmann H, Choi H | 2020 | Remodelling of whole-body lipid metabolism and a diabetic-like phenotype caused by loss of hepatic CDK1 and hepatocyte division | https://www.ncbi.nlm.nih.gov/geo/query/acc.cgi?acc=GSE159498 | NCBI Gene Expression Omnibus, GSE159498 |

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
