## [Decision Letter]

[Editors' note: this paper was reviewed by Review Commons.]

**Acceptance summary:**

NAFLD while of profound physiological significance to human health remains somewhat mysterious in terms of its cause, which limits new therapeutic approaches. This paper links hepatocyte proliferation to dysregulation of lipid metabolism and proposes a new mechanism to explain the development of age-dependent NAFLD. It is an important paper for the field and should be of widespread interest- particularly researchers in the fields of liver disease, ageing and lipid metabolism.

---

## [Author Response]

Your study provides impressive evidence linking CDK1 to hepatic steatosis and liver disease. All reviewers were highly impressed by the diverse array of techniques and approaches that you brought to bear on this interesting problem. I have read the reviewers comments and your response to each of these and it is my opinion that you should be able to respond to these comments within a reasonable period of time. Some of the points raised by Rev #2 are particularly important and you should try to address these in detail. One point that I felt was important to address was the issue of whether CDK1 is only involved in cell proliferation or does it control other aspects of liver function. This may require some softening of some of the claims and conclusions made throughout.

Thank you for these constructive comments. We have followed your and the reviewers’ suggestions and have addressed all the points that were raised. We added new data in Figure1I, 1J, 1L, 2K, and 9. In addition, we toned down several of our claims by providing alternative explanations. Overall, we have improved this revised version of our manuscript and hope it can be published in its current form.

Reviewer #1:In liver disease like NAFLD or NASH, it is known that proliferation of hepatocytes is impaired. This is thought to be caused by lipid droplet accumulation in hepatocytes and therefore being a secondary effect. The decrease in hepatocyte proliferation is an aggravating factor in the disease since the repair of liver damage relies on self-renewal of hepatocytes. In this manuscript the authors investigate if blocking hepatocyte self-renewal (by knocking out CDK1) has any effects on lipid metabolism. Partly this is based on their previous work where they studied glucose metabolism. The authors find reduced triacylglycerides (TGs) in young animals, which is caused by oxidative stress activating the transcription factor FOXO1 to drive the expression of Pnpla2 (encoding the ATGL protein). They ChIP FOXO1 to the Pnpla2 promoter but more importantly silencing of Pnpla2 in the liver of mice, restores TGs to wild type levels. In addition, reducing oxidative stress using NAC has the same outcome than silencing Pnpla2. Furthermore, the authors detected impaired β-oxidation, reflected by increased acylcarnitines and reduced β-hydroxybutyrate, which led to high levels of free fatty acids in plasma that was transported to and stored in adipose tissue. Eventually, this increased insulin secretion. When the mice were aged, the hyperinsulinemia caused insulin resistance and hepatic steatosis, similarly as is observed in diabetic patients.This is a highly significant manuscript with a lot of valuable data, which will be useful for a broad audience. The work presented here is a tour-de-force, where the authors use systems biology including animals, RNA sequencing, lipidomics, metabolomics, ChIP, histology, physiology, and biochemical approaches. It is also important that not only the liver was studied but also adipose tissue and skeletal muscle in addition to blood parameters. Overall, this study is well-rounded, investigates an important biological problem with clinical implications, and used state-of-the-art approaches.The first important finding is the axis of loss of hepatocyte self-renewal/Cdk1->oxidative stress->FOXO1->Pnpla2. Especially since both NAC and silencing of Pnpla2 lead to the restoration of TGs. The second important point is how the impairment of hepatocytes affects other organs like adipose tissue and skeletal muscle. The authors have done a good job to provide a complete picture of the biology how these tissues interact in terms of lipid metabolism. The final important point is the investigations of the age mice both at 6 and 12 months, which provides a picture how the disease progresses and this resembles much more what happens in patients in the clinic.In summary, the authors provide strong evidence that the loss of hepatocyte self-renewal causes changes in lipid metabolism that resemble those seen in patients with liver disease or diabetes.There are a number of issues, which need to be addressed:1) In the ChIP experiments (Figure 1M and 7G) some important negative controls are missing. The authors need to add these.

We have included the IgG isotype controls in the figures for the ChIP experiments (Figure 10 and Figure 7G) to prove that our experiments are well controlled.

2) Although the authors report the number of up/down-regulated genes in the RNA sequencing experiments, the total number of genes detected have not been reported.3) In the lipidomics experiments, the authors state "The majority of lipid species were unchanged" but they need to mention how many lipid species they were able to detect reliably. This is important information.

We agree with point 2 and 3 and have incorporated into the text the total number of detected genes from the RNAseq and detected lipid species from the lipidomics data. We have also added the complete list of genes detected in Supplementary file 6, in addition to the list of differentially expressed genes that was already in Supplementary file 6.

4) Can the authors explain how oxidative stress induces FOXO1 in hepatocytes?

We have provided mechanisms on how oxidative stress might induce FOXO1 activity in the Discussion section.

5) The authors should speculate how it can come to a reversal of the phenotype when comparing young and old mice.

We thank the reviewer for bringing up this point. To be more explicit, we have described how we believe the development of insulin resistance can cause the reversal of hepatic lipid phenotype upon aging in the Discussion section.

6) Adiponectins are mentioned (Figure 5—figure supplement 1E) but how they contribute to the phenotype needs to be explained more carefully.

We have included a short description on how adiponectin may mechanistically contribute to insulin resistance in the text.

7) Using metabolomics, the authors identify the purine metabolism pathway but this finding needs to be explained more carefully and these findings need to be put in context.

We have provided a paragraph of Discussion on why purine metabolites might be increased in insulin resistance, especially in the context of the *Cdk1* cKO mice.

Significance:This manuscript is of high significance due to its findings, its investigations of the molecular details, and the number of unbiased approaches used. It is an important finding that impairment of hepatocyte self-renewal leads to remodeling of lipid metabolism since the opposite was expected. For many years, lipid droplet accumulation in hepatocytes was thought to be the reason for impaired self-renewal. Therefore, it is possible that lipid droplet accumulation is not causally linked to hepatocyte self-renewal but this will need to be further investigated. In my opinion, this manuscript should be published in a high-quality journal.

We thank this reviewer for the support of our manuscript.

Reviewer #2:Authors propose that loss of hepatocyte proliferation (achieved through specific deletion of CDK1 in hepatocytes) results in changes in lipid metabolism and with aging contributes to insulin resistance and hepatic steatosis. The authors show that CDK1-/- hepatocytes contain decreased level of triacylglycerides and decreased fatty acid oxidation- which results in elevated storage of FFA as triacylglycerides in WTA.Major comments:Data is of very high quality, well designed and performed and presented clearly. Authors used multiple approaches to investigate their hypotheses- including omics approaches (lipidomics, RNA-seq), molecular biology approaches and physiological assessments. In my view, the key conclusions are supported by the evidence and the model presented is plausible. I have some general comments- which in my view do not require extensive additional experimentation- but that the authors should consider if possible to address experimentally and/or discuss.I wonder if hepatocyte proliferation is really an important factor here- hepatocyte proliferation occurs at very low frequency in uninjured liver (could the lipid metabolism phenotypes observed due to loss of CDK1 be uncoupled from the role of CDK1 in hepatocyte proliferation?).

This is an important point that the reviewer has brought up. We agree that the observed phenotypes might be due to a specific loss of CDK1 activity. As such, we have sought to bring forward the limitation of the *Cdk1* cKO mouse model as a generalization of the loss of hepatocyte proliferation in the Discussion. In general, we have toned this down and provided several alternative explanations.

Aged CDK1-/- mice (12 months old) showed increased blood glucose, insulin resistance, hepatic steatosis and liver fibrosis, which is a very interesting observation- I wonder if the authors could comment on the physiological relevance. Is Cdk1 expression decreased during natural aging in hepatocytes? Does it impact on hepatocyte proliferation during the aging process-in the absence of liver injury?

We have discussed how CDK1 activity is reduced during aging, both from reduced transcriptional induction and increased senescence, which makes aging a physiologically relevant context whereby findings from *Cdk1* cKO mice are applicable. As the reviewer has brought up, we believe it is also in part responsible for reduced proliferative capacity of aged hepatocytes. However, it is quite unlikely that aging occurs in the absence of accumulated liver injury, as the liver is constantly exposed to toxins for clearance, resulting in hepatocyte death and consistent need for hepatocytes to divide for homeostatic renewal.

IS decreased Cdk1 a feature of NASH?

Meta-analysis of transcriptomics data from NAFLD patient livers does not find a difference in CDK1 transcript levels, but we have also highlighted that senescence markers are higher in NAFLD liver, in which case Cdk1 activity is inhibited.

Authors show increased steatosis and fibrosis markers in aged CDK1-/- livers which are characteristic of NASH- what about inflammatory markers such as expression of pro-inflammatory genes and immune cell infiltrates?

We have highlighted the absence of immune infiltrates in the aged *Cdk1* cKO livers and have proposed that insulin resistance in the aged *Cdk1* cKO mice be immunosuppressive and prevent manifestation of inflammation in the aged liver.

As mentioned in the Introduction- there is plenty of evidence that hepatocyte cellular senescence is (i) observed during NAFLD; (ii) contributes causally to steatosis. Does deletion of CDK1 contribute to hepatocyte senescence?

We have performed senescence-associated β-galactosidase staining as well as qPCR analysis of senescent markers, both of which indicate increased presence of senescent cells in *Cdk1* cKO livers (Figure 9). As such, we have highlighted this as a potential limitation of the *Cdk1* cKO model.

Authors should cite very recent paper- by Omori S and colleagues (Cell metabolism)- which shows that elimination of p16ink4a cells ameliorates steatosis and inflammation in a NASH model.

We have cited the paper by Omori *et al.* in our Discussion on the involvement of senescence in the NAFLD phenotype.

Minor commentsText and Figures are generally clear and accurate. In Figure 1 A- heat map of hepatic triacylglycerides should indicate the species otherwise it is not particularly informative.

We have included the lipid species in Figure 1A.

The fact that anti-oxidant enzymes are up-regulated at the mRNA level in CDK1-/- is evidence of increased oxidative stress (may be the opposite)- however, the carbonylation data is convincing in indicating increased oxidative stress.

We agree that the mRNA expression of oxidative stress genes can be interpreted both ways and therefore the protein carbonylation data is an important factor.

SignificanceThis paper links hepatocyte proliferation to dysregulation of lipid metabolism- and proposes a new mechanism to explain the development of age-dependent NAFLD. It is an important paper for the field and should be of widespread interest- particularly researchers in the fields of liver disease, aging and lipid metabolism.I am expert in aging and age-related disease and the role of aging in lipid metabolism.Reviewer’s cross-commentingI have no additional comments. All points raised are valid and I am generally in agreement with the assessment from the other reviewers. This study is well designed and its findings significant.

We thank this reviewer for the constructive comments and the support of our manuscript.

Reviewer #3:Cell proliferation and cell metabolism are intimately linked with well-described regulatory pathways involved in the coordinated control of these processes. As but one example, in order to ensure an adequate supply of essential building blocks required for membrane biosynthesis, changes in lipid uptake/synthesis/storage accompany proliferation. In the current study, Ow and colleagues take advantage of a mouse model of impaired hepatocyte proliferation (Cdk1Liv-/- mouse) to thoroughly examine the impact of proliferation on lipid metabolism and systemic physiology. In young mice, the authors report defects in lipolysis and fatty acid oxidation that increase adiposity and enhance insulin secretion. In aged mice, insulin resistance and hepatic steatosis is observed. In light of these results, the authors propose a previously uncharacterised role for impaired hepatocyte proliferation in liver disease.This is an elegant study with a logical flow and high-quality data to support the hypotheses and conclusions. All experiments are adequately replicated and statistical analysis is appropriate. The Materials and methods are thorough, and all experiments could be reproduced by others in the field. As outlined below there are some minor experiments, that are neither time nor resource heavy, that would improve the study.The authors conclude that the reduction in TGs in hepatocytes from Cdk1Liv-/- mice results from increased transcription of Pnpla2/ATGL. Ideally, in Figure 1K, the authors should demonstrate that ATGL protein expression is increased in Cdk1Liv-/- liver lysates. This will complement the mRNA expression data shown in Figure 1H.

We have performed immunoblot for ATGL on 8-week-old liver lysates and have found that ATGL protein level is increased, supporting the mRNA data. The data has been incorporated as Figure 1I-J.

In relation to Figure 2I/J, the authors refer to a study showing that CDK1 phosphorylates SIRT3 to promote CPT2 dimerization and fatty acid oxidation (Liu et al., 2020). Clearly SIRT3 phosphorylation is dramatically reduced in the CdkLiv-/- setting but the authors should also demonstrate changes in CPT2 acetylation or CPT2 enzyme activity in their model.

We have immunoprecipitated CPT2 and ran an immunoblot, probing for pan-acetylation. In agreement with our hypothesis that SIRT3 activity is reduced, there is an increase in acetylated residues on CPT2. The data is shown now in Figure 2K.

In Figure 7B, the authors demonstrate decreased expression of Foxq1 in Cdk1Liv-/- samples from aged mice. Given the association between FOXQ1 and FOXO1, is there any difference in FOXQ1 transcript or protein expression in Cdk1Liv-/- samples from young mice?

We have performed qPCR for *Foxq1* and do not observe significant differences at the mRNA level. The data has been included in Figure 1L. We also discuss that Foxq1 levels are different in young and old *Cdk1* cKO mice.

The authors should take care when drawing conclusions about the impact of impaired hepatocyte proliferation on liver disease. The authors have clearly shown that complete loss of hepatic CDK1, and therefore induction of a proliferative defect in all hepatocytes, remodels lipid metabolism, promotes a diabetic-like phenotype and contributes to liver disease. Senescence, or other proliferative defects, will occur in a more mosaic fashion and might therefore not trigger all of the biology observed in the Cdk1Liv-/- mice. The authors should include a discussion of this potential limitation.

We totally agree and have discussed this by highlighting this limitation of the *Cdk1* cKO mice in the Discussion.

SignificanceThis study represents a conceptual advance and provides convincing evidence linking changes in hepatocyte proliferation, and ensuing metabolic reprogramming events, with disturbed systemic metabolism. The idea that altered hepatocyte proliferation directly contributes to liver disease is intriguing and this study certainly opens multiple angles for future research. It is a nice complement to work from Ogrodnik and colleagues (Nature Communications 8:15961, 2017) showing that defective FAO in senescent hepatocytes contributes to hepatic steatosis and provides important insights regarding the relationship between cellular metabolism, systemic metabolism and metabolic disease. This study will be of interest to a diverse audience including those interested in liver biology, senescence, cellular metabolism, systemic metabolism and metabolic disease. My area of research is cellular metabolism with a particular focus on cancer cell metabolism. I am involved in collaborative projects to investigate hepatocyte metabolism during development, regeneration and tumorigenesis so believe that I have sufficient expertise to evaluate all aspects of this study.Reviewer's cross-commentingI have no additional comments. There was clearly an overarching consensus amongst the reviewers regarding the quality of the study and impact of the findings. This is an important study that will be of interest to a diverse audience.

We thank this reviewer for the constructive comments and the support of our manuscript.